# Binding of the periplakin linker requires vimentin acidic residues D176 and E187

Elena Odintsova[1,8], Fiyaz Mohammed[2,8], Catharine Trieber [3,8], Penelope Rodriguez-Zamora [1,6], Caezar Al-Jassar[1], Tzu-Han Huang[4], Claudia Fogl[1,7], Timothy Knowles [4], Pooja Sridhar[4], Jitendra Kumar[3], Mark Jeeves[1], Martyn Chidgey[1,5✉] & Michael Overduin [3]

Plakin proteins form connections that link the cell membrane to the intermediate filament cytoskeleton. Their interactions are mediated by a highly conserved linker domain through an unresolved mechanism. Here analysis of the human periplakin linker domain structure reveals a bi-lobed module transected by an electropositive groove. Key basic residues within the periplakin groove are vital for co-localization with vimentin in human cells and compromise direct binding which also requires acidic residues D176 and E187 in vimentin. We propose a model whereby basic periplakin linker domain residues recognize acidic vimentin side chains and form a complementary binding groove. The model is shared amongst diverse linker domains and can be used to investigate the effects of pathogenic mutations in the desmoplakin linker associated with arrhythmogenic right ventricular cardiomyopathy. Linker modules either act solely or collaborate with adjacent plakin repeat domains to create strong and adaptable tethering within epithelia and cardiac muscle.

[1] Institute of Cancer and Genomic Sciences, University of Birmingham, Birmingham B15 2TT, UK. [2] Institute of Immunology and Immunotherapy, University of Birmingham, Birmingham B15 2TT, UK. [3] Department of Biochemistry, Faculty of Medicine & Dentistry, 474 Medical Sciences Building, University of Alberta, Edmonton, Alberta T6G 2H7, Canada. [4] School of Biosciences, University of Birmingham, Birmingham B15 2TT, UK. [5] Institute of Clinical Sciences, University of Birmingham, Birmingham B15 2TT, UK. [6] Present address: Instituto de Fisica, Universidad Nacional Autonoma de Mexico, Mexico City 04510, Mexico. [7] Present address: The Binding Site, Birmingham B15 1QT, UK. [8] These authors contributed equally: Elena Odintsova, Fiyaz Mohammed, Catharine Trieber. ✉email: M.A.Chidgey@bham.ac.uk

 

The plakin proteins connect the three elements of the cytoskeleton, namely intermediate filaments (IFs), micro-filaments and microtubules, to each other, to junctional complexes at the membrane and to intracellular organelles. There are seven members of the superfamily in mammals: peri-plakin, envoplakin, desmoplakin, plectin, bullous pemphigoid antigen 1 (BPAG1; also known as dystonin), epiplakin and microtubule–actin cross-linking factor 1. The roles of family members in diverse biological processes, including cell–cell and cell–matrix adhesion, cell migration, mechanotransduction and cell signalling[1], are critically dependent upon their ability to interact with the cell cytoskeleton. Plakin protein recognition of IFs is mediated by plakin repeat domains (PRDs) and linker modules. The former interact with IF proteins via complementary electrostatic interactions[2,3], but the molecular mechanism by which linker modules connect to the IF cytoskeleton remains elusive.

Periplakin and envoplakin initiate formation of the cornified envelope, a layer of cross-linked protein that forms beneath the plasma membrane during keratinocyte differentiation, creating the skin's permeability barrier[4]. These proteins are targeted by antibodies in paraneoplastic pemphigus, a mucocutaneous skin blistering disorder that accompanies neoplasia, often via C-terminal linker-containing sites[5]. Such antibodies disrupt keratinocyte cell adhesion in culture, although the mechanism under-lying this effect remains obscure[6]. Desmoplakin is a constituent of desmosomes that form strong junctions between cells in epithelia and cardiac muscle. It bridges the gap between other desmosomal proteins and keratin IFs in epithelial cells, desmin IFs in cardiomyocytes and vimentin IFs in meningeal cells and follicular dendritic cells of lymph nodes[7]. Mutations in desmoplakin result in an array of diseases that affect the skin, hair and heart and sometimes all three[8]. Arrhythmogenic right ventricular cardiomyopathy (ARVC) leads to cardiac arrest and sudden death and results from mutations in the genes encoding desmosomal proteins expressed in the heart[9]. Pathogenic mutations are dispersed throughout desmoplakin, including in the C-terminal tail region responsible for engaging IFs[10].

Plectin is expressed in skin, muscle and peripheral nerve, and links the IF cytoskeleton to hemidesmosomal cell–matrix junctions in the epidermis, and to various structures in skeletal, smooth and cardiac muscle[11]. Mutations in plectin cause the skin blistering disease epidermolysis bullosa simplex (EBS)[1] and limb-girdle muscular dystrophy[12]. The BPAG1e is isoform expressed in the epidermis, interacts with IFs and contributes to the structural integrity of hemidesmosomes[13]. Mutations in DNA encoding BPAG1e cause EBS[14], and circulating anti-BPAG1 antibodies are detected in patients with the autoimmune skin blistering disease bullous pemphigoid[15].

Common to all these proteins is the conserved linker domain, which lacks a validated structure and mechanism. The periplakin linker module sequence comprises 110 residues, encompassing most of periplakin's conserved C-terminal tail region, and is solely responsible for direct IF tethering, underscoring its functional significance. The C-terminal tails of other plakin proteins, including envoplakin, desmoplakin, plectin and BPAG1e also contain a linker module as well as a series of PRDs comprised of a number of PR modules. Envoplakin has just one PRD, while BPAG1e, desmoplakin and plectin contain two, three and six, respectively. PRDs are globular modules that possess a basic binding groove that accommodates IF rods through complementary electrostatic interactions[2,3]. In envoplakin the linker domain joins its central rod to its singular PRD whereas in desmoplakin, BPAG1e and plectin the linker connects the penultimate and C-terminal PRDs, suggesting functional interconnectivity. The interaction of PRDs and linkers with IFs is vital for the

maintenance of tissue integrity. Truncating mutations that result in the loss of all three desmoplakin PRDs and the linker region cause lethal acantholytic epidermolysis bullosa, a devastating skin blistering disease that is characterised by catastrophic fluid loss and early death[16].

The periplakin linker domain is unusual in that it constitutes the only means by which periplakin can directly interact with IFs. It interacts with keratin 8 and vimentin in yeast two-hybrid and protein–protein interaction assays[17], and when transfected into cultured cells it co-localises with IFs[17–19]. A crystal structure of a periplakin linker construct has been determined (PDB entry 4Q28). It displays an elongated shape that fits into a molecular envelope of desmoplakin's C-terminus[20]. The PR-like motif structure closely resembles the canonical PR2 repeat[2], with the notable exception that the second helix (H2) is shorter in periplakin. The larger C-terminal PR-like module within the periplakin linker aligns well with the N-terminal (Nt) PR-like motifs found in desmoplakin PRD-B and PRD-C modules[2]. A peculiar feature of the periplakin linker structure is that a N-terminal hexa-histidine tag forms an extended β strand that pairs with the corresponding region of a neighbouring symmetry related molecule in the crystal lattice. This packing arrangement of the affinity tag into the linker fold is clearly non-physiological. Moreover, the crystallised periplakin construct lacks conserved N-terminal residues that could normally form part of the structure. Due to these artifacts, structural and functional validation is needed. Herein, we identify a basic groove within the periplakin and desmoplakin linkers, and show that mutations within their grooves disrupt co-localisation with vimentin IFs in transfected cells. We also identify residues in periplakin linker and vimentin that are critical for the interaction between the two proteins, and propose a mechanism for recognition of IF binding motifs.

## Results

**The periplakin linker reveals a positively charged groove.** The periplakin linker is found at the extreme C-terminus of the periplakin protein (Supplementary Fig. 1a). The crystal structure of the periplakin linker was determined by the Northeast Structural Genomics Consortium (PDB entry 4Q28) and briefly described by Weis and colleagues[20]. In an attempt to verify the crystal structure and resolve issues arising from the hexahistidine tag, and the lack of N-terminal residues (K1646–L1654) that are relatively conserved across the plakin family and highly conserved in periplakins from different species (Supplementary Fig. 2a, b), we attempted to determine the solution structure by nuclear magnetic resonance (NMR). This was precluded by the lack of stability of the protein despite extensive optimisation of solution conditions. As an alternative we generated an I-TASSER model of the periplakin linker encompassing residues K1646–K1756 (Fig. 1a). The model displays a similar secondary topology to that of the crystal structure, forming a bi-lobed module connected by long β-strands, although in the I-TASSER derived model the dihedral angles for H1653 deviate from a typical β-sheet con-formations leading to a disruption in the strand at the extreme N-terminal end. Support for the model comes from the HSQC spectrum of the $^1$H$^{15}$N-labelled periplakin linker which is well-dispersed, as well as circular dichroism data that demonstrate that the protein adopts an α/β fold structure (Fig. 1b, Supplementary Fig. 2c). In addition secondary structure prediction for the desmoplakin linker based on NMR chemical shift data and calculated using Talos$^+$ suggest that the desmoplakin linker has a central β-strand in solution (Supplementary Fig. 2a). Given the high sequence similarity between the desmoplakin and periplakin linkers it is likely that the periplakin linker adopts a similar

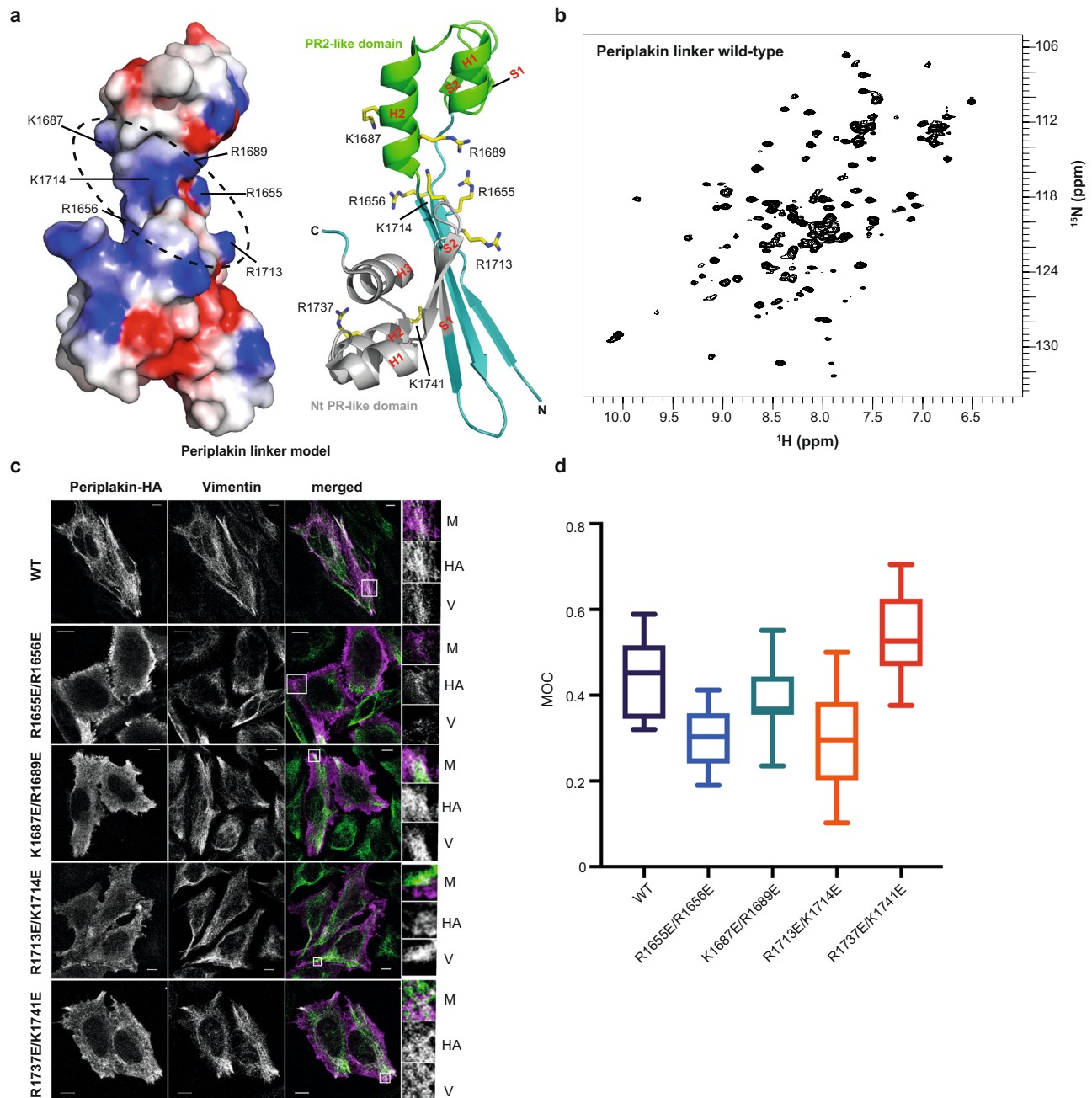

**Fig. 1 Co-localisation with vimentin intermediate filaments is compromised by mutations in periplakin's basic groove. a** Electrostatic surface potential and ribbon representation for the I-TASSER derived periplakin linker domain model (C-score = 0.99). Electrostatic surface potential was calculated with DelPhi with the potential scale ranging from −7 (red) to +7 (blue) in units of $kT/e$. The putative vimentin docking site is shown (black dashed circle). The ribbon model shows a PR2-like motif (green) and an Nt PR-like module (grey), and highlights the position of residues (stick format) mutated in periplakin transfected cells. **b** Two dimensional $^{1}$H,$^{15}$N-resolved NMR spectra of the periplakin linker domain. **c** Constructs encoding residues M1588–K1756 of human periplakin with a C-terminal HA tag were transfected into HeLa cells. Cells were stained with anti-HA and anti-vimentin antibodies. Periplakin is stained purple and vimentin is stained green in the merged images. The boxed areas are expanded on the far right-hand side. M merged image, HA periplakin staining, V vimentin staining. Bars, 10 μm. **d** Manders' overlap coefficient (MOC) was calculated for each image and is shown as a Tukey box plot with the median and 25th and 75th percentiles of each distribution. An unpaired $t$ test with Welch's correction was performed on the data: Wild-type (WT) versus R1655E/R1656E, $p = 0.002$; WT versus K1687E/R1689E, not significant; WT versus R1713E/K1714E, $p = 0.03$; WT versus R1737E/K1741E, $p = 0.03$. At least five fields of view were analysed for each experiment. z-stacks were taken for each field and overlap coefficients calculated for each individual z-stack. An average overlap coefficient was then calculated for each experiment and each experiment was repeated two to three times.

overall topology in solution. The precise conformation of the extreme N-terminus and first β-strand of periplakin's linker merits further experimental analysis. Nevertheless the structure does unequivocally reveal two PR-like motifs that flank a central basic groove that could accommodate a IF rod (Fig. 1a). To

identify candidate periplakin residues responsible for IF recognition detailed analysis of the central basic groove was performed. The groove is enriched with positively charged residues including R1655, R1656, K1687, R1689, R1713 and K1714 (Fig. 1a), several of which are highly conserved (Supplementary Fig. 2a, b) based

on primary sequence analysis with the PRALINE (PRofile ALIgNEment) tool[21].

**Basic groove mutations compromise interaction with vimentin.** The function of the linker domain's basic groove was investigated by testing the effects of mutations on the localisation of a periplakin construct to IFs in transfected HeLa cells. The periplakin construct consisted of a C-terminal portion of the rod domain, the linker domain and a haemagglutinin antigen (HA) tag (Supplementary Fig. 3a). This construct has previously been shown to co-localise with IFs in transfected cells[3,22], and as expected showed extensive co-localisation with vimentin IFs when transfected into HeLa cells with HA staining matching filamentous staining for endogenous vimentin (Fig. 1c). In order to perturb IF recognition, a series of double charge reversal mutations were designed within the periplakin linker groove. In particular the surface exposed R1655, R1656, K1687, R1689, R1713 and K1714 residues were changed to glutamates. For a control R1737 and K1741, which protrude from the Nt PR-like module and are outside the groove, were substituted. Double mutants R1655E/R1656E and R1713E/K1714E showed similar patterns of staining with both mutant periplakin proteins mainly distributed in small aggregates at the cell periphery (Fig. 1c). Double mutant K1687E/R1689E showed a diffuse staining pattern throughout the cytoplasm. In all three cases the pattern of staining was strikingly different from that of the wild-type periplakin protein. By contrast, double mutant, R1737E/K1741E demonstrated a comparable pattern of staining to the wild-type construct. Expression of wild-type and mutant periplakin constructs was similar by western blotting (Supplementary Fig. 3b). Together this indicated that co-localisation was specifically compromised by charge reversal mutations inside the putative binding groove.

Two approaches were used to confirm the importance of residues R1655, R1656, K1687, R1689, R1713 and K1714 in IF binding. Quantification of co-localisation between periplakin proteins and vimentin in transfected HeLa cells was analysed using the Manders' method[23], which measures the fraction of pixels with positive values in two channels. The values of Manders' overlap coefficient (MOC) range from 0 to 1 with an overlap coefficient of 0.5 implying that one protein (as a fraction of the fluorescence in one channel) co-localises with 50% of a second protein in another channel. Cells transfected with the wild-type periplakin construct exhibited an average MOC of ~0.45, indicating substantial co-localisation with vimentin (Fig. 1d). Co-localisation was reduced in cells transfected with double mutants R1655E/R1656E and R1713E/K1714E, with average MOC values of 0.30 and 0.38, respectively. In cells transfected with double mutant K1687E/R1689E the MOC was ~0.43, which is similar to that observed in wild-type cells, presumably based in part on the diffuse cytosolic distribution of the delocalised periplakin seen in these cells. The R1737E/K1741E control cells displayed a MOC value of ~0.57, indicating preservation of vimentin IF co-localisation.

To confirm the direct nature of the periplakin–vimentin interactions, in vitro binding experiments were performed. A periplakin construct spanning the entirety of the linker domain's conserved sequence (K1646–K1756) was purified to homogeneity. Binding of the periplakin linker domain to a vimentin$^{ROD}$ protein encompassing coils 1A and 1B of the central rod domain (residues T99–I249; Supplementary Fig. 1b) was measured. The vimentin$^{ROD}$ was labelled with a NT647 fluorescent group and incubated with increasing concentrations of the periplakin linker domain in the presence of 150 mM NaCl for microscale thermophoresis (MST)-based binding assays (Fig. 2a, Supplementary Table 1). Wild-type

periplakin linker bound to vimentin$^{ROD}$ with a $K_D$ of 70.5 ± 3.8 μM. Periplakin linker proteins containing mutations R1655E/R1656E, K1687E/R1689E, R1713E/K1714E and R1737E/K1741E were purified to homogeneity. All mutant proteins were folded, as indicated by the similarity of their $^1$H,$^{15}$N resolved spectra to that of the wild-type protein (Supplementary Fig. 4a). Notably, variants R1655E/R1656E, K1687E/R1689E and R1713E/K1714E showed compromised binding to vimentin$^{ROD}$ based on their affinities of 380 ± 51, 300 ± 54 and 135 ± 30 μM, respectively (Fig. 2a, Supplementary Fig. 4b). Binding of control double mutant R1737E/K1741E to vimentin$^{ROD}$ was slightly stronger than that of the wild-type protein ($K_D = 48 ± 11$ μM versus 70.5 ± 3.8 μM). Interestingly, this mutant also displayed a higher MOC value than the wild-type protein (Fig. 1d). Collectively, these results support an electrostatic mode of interaction in which basic residues R1655, R1656, K1687, R1689, R1713 and K1714 within periplakin's binding groove (Fig. 1a) recognise vimentin filaments. To explore the role of electrostatics in the binding, we examined the effect of salt on the interaction (Supplementary Fig. 5, Supplementary Table 1). Decreasing the salt concentration from 150 to 10 mM NaCl led to enhanced affinity of the wild-type linker/vimentin interaction from 70.5 ± 3.8 to 31 ± 2 μM, indicating that electrostatic attraction plays a role in linker domain-IF binding.

A critical motif for IF targeting and co-localisation in transfected cells has previously been mapped to periplakin linker residues 1694–1698 (DWEEI) based on deletion studies[22]. We mapped this motif onto the periplakin linker domain model (Fig. 2b). This highly conserved element is located in the PR2-like motif and is proximal to the basic groove. The carboxyl group of D1694 mediates an ionic interaction with R1655, a residue that has proved to be critical for vimentin binding. Furthermore, E1696 forms a salt bridge interaction with R1713 and this interaction may allow R1713 to adopt a conformation that favours IF binding (Fig. 2b). Finally, W1695 is also situated in close proximity to the putative IF binding groove and forms extensive non-polar stacking interactions with the residues emanating from the extreme helix (H3) of the Nt PR-like motif (Fig. 2b), thereby stabilising this region. Strikingly, mutation of residues 4274–4277 (RKRR) in the plectin linker (equivalent to periplakin residues L1654–S1657) to ANAA also abolishes IF co-localisation in transfected cells[24]. Mapping of these basic residues onto the I-TASSER derived plectin linker model (K4266-A4377) reveals that K4275 and R4277 line the groove (Fig. 2c). Taken together this supports the role of the basic groove as a key IF recognition determinant.

**Desmoplakin linker domain mutations affect co-localisation.** To investigate the IF binding mechanism of the desmoplakin linker a model (residues Q2454–N2565) was calculated using I-TASSER (Fig. 3a). The desmoplakin linker exhibited a basic groove lined with positively charged side chains that included residues K2463, R2464, K2494, R2522 and K2523 (Fig. 3a). The desmoplakin linker domain expressed poorly in transfected HeLa cells, necessitating use of a larger desmoplakin construct (DSP$^C$, residues T1960–A2822) which encompasses all three PRDs and the linker domain (Supplementary Fig. 3a). Similar constructs co-localise with vimentin IFs in cultured cells[25–27]. Following transfection into cultured HeLa cells the DSP$^C$ protein co-localised with vimentin IFs (Fig. 3b) with a MOC of 0.5 (Fig. 3c). To determine the role of the desmoplakin linker domain in vimentin targeting we deleted it from the DSP$^C$ construct to produce a truncated protein DSP$^C$ΔLinker. Although the MOC for DSP$^C$ΔLinker only showed a small reduction when compared to the wild-type protein (Fig. 3c) there was a dramatic difference in staining pattern with DSP$^C$ΔLinker distributed in dense

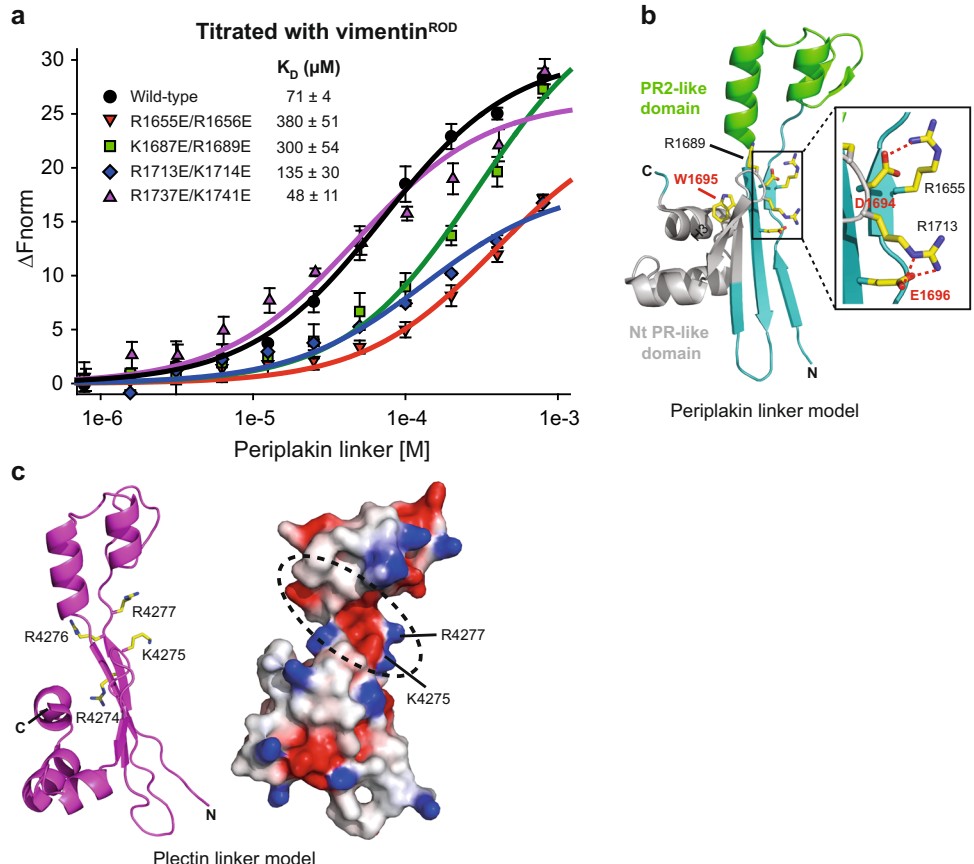

**Fig. 2 Mutations in the basic groove of the periplakin linker affect binding to the vimentin$^{ROD}$. a** Increasing concentrations of periplakin linker domains (0.78 μM to 800 μM) were incubated with 50 nM labelled vimentin$^{ROD}$ protein and interactions were measured by MST. The baseline-corrected normalised fluorescence Δ$F_{norm}$ is plotted against increasing concentration of the linker domain. Data points are mean ± SD from three to four independent experiments and were fit to a one-site binding model. The binding affinity, $K_D$, was estimated from these fits to be WT = 71 ± 4 μM, R1655E/R1656E = 380 ± 51 μM, K1687E/R1689E = 300 ± 54 μM, R1713E/K1714E = 135 ± 30 μM, and R1737E/K1741E = 48 ± 11 μM. **b** Ribbon representation of the periplakin linker model domain highlighting residues 1694 DWE 1696 (labelled in red). Residue W1695 forms stacking interactions with helix 3 (H3) in the Nt PR-like motif. The box shows a close-up view of the ionic interactions mediated between D1694 and R1655 and E1696 and R1713 (red dashed lines). **c** Ribbon representation and electrostatic surface potential for the plectin linker domain structural model derived from I-TASSER (C-score = 0.64). Residues 4274 RKRR 4277 are shown (stick format). A putative IF docking site in the plectin linker is highlighted (black dashed circle).

dot-like structures, predominantly in the perinuclear area (Fig. 3b). When glutamate substitutions of desmoplakin residues K2463 and R2464 (equivalent to periplakin groove residues R1655 and R1656) were introduced into the DSP$^C$ construct staining was concentrated predominantly at the cell periphery and the IF co-localisation was significantly reduced with a MOC of ~0.36 (Fig. 3b, c). This suggests that the basic groove in the desmoplakin linker also contributes to targeting the protein to the cytoskeleton and thus constitutes the consensus function of this module.

Residues C2501 and E2502 within the desmoplakin linker domain are deleted in the ARVC mutant C2501-E2502del (Supplementary Fig. 1a)[28]. Deletion of these residues resulted in a staining pattern similar to that obtained with the DSP$^C$ΔLinker protein, i.e. dense dot-like structures predominantly at the cell periphery, with a corresponding reduction in the MOC (Fig. 3c). To examine the effect of the C2501-E2502del mutation on linker domain structure we collected a $^1$H,$^{15}$N-HSQC NMR spectrum of a linker domain construct lacking these two residues, and found only minor signal perturbations (Fig. 4a). Aside from those residues directly adjacent to the mutation the majority of the peak differences between the spectra of the wild-type desmoplakin linker and the C2501-E2502del mutant map to residues within the α-helices of the Nt PR-like motif, suggesting slight

conformational changes in this region (Fig. 4b). Examination of the putative desmoplakin linker structure showed that E2502 mediates salt bridge interactions with K2463 and R2464 (Fig. 4c), and it is likely that it holds these two residues in a conformation that facilitates IF binding. A model of the desmoplakin linker with the C2501-E2502del mutation was generated to illuminate this issue. Although deletion of residues C2501 and E2502 is unlikely to severely compromise the overall secondary structure arrangements relative to wild-type desmoplakin linker, subtle differences within the positive groove were found. In the absence of E2502 the R2464 side chain is predicted to swing away from the IF binding groove region. However, the nearby carboxylate group of E2503 may form compensatory salt bridge interactions with R2522 and K2463 (Fig. 4d). It is conceivable that these rearrangements result in the partial loss of co-localisation with vimentin seen in the transfection experiments (Fig. 3b, c).

Introduction of the ARVC mutation R2541K (Supplementary Fig. 1a)[29] into the desmoplakin linker domain showed a similar effect to the K2463E/R2464E mutant. That is, the mutant distributed predominantly to the plasma membrane and exhibited a reduced MOC (Fig. 3b, c). Again, only minor alterations in the $^1$H,$^{15}$N NMR spectra were observed when compared to the wild-type linker protein (Supplementary Fig. 6a). The majority of residues exhibiting the largest chemical shift

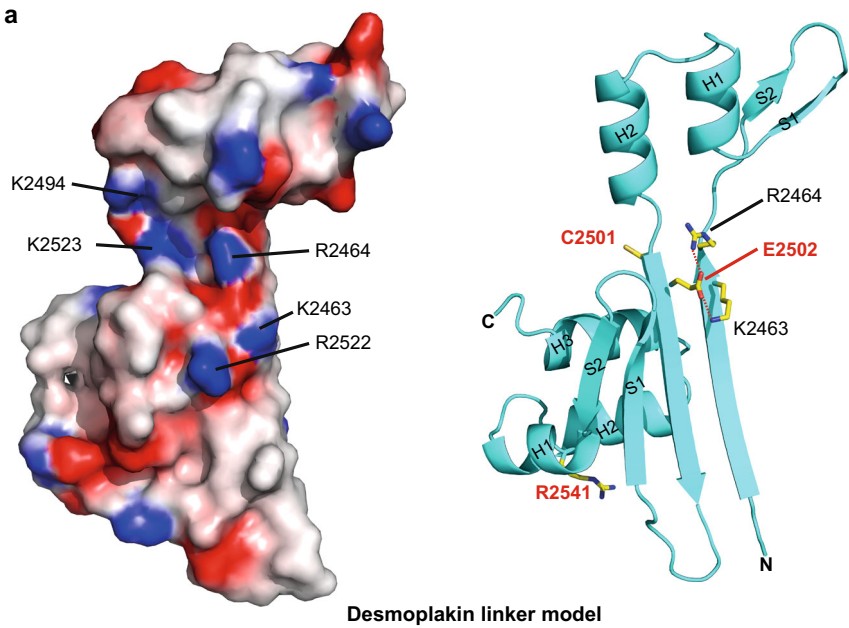

**Desmoplakin linker model**

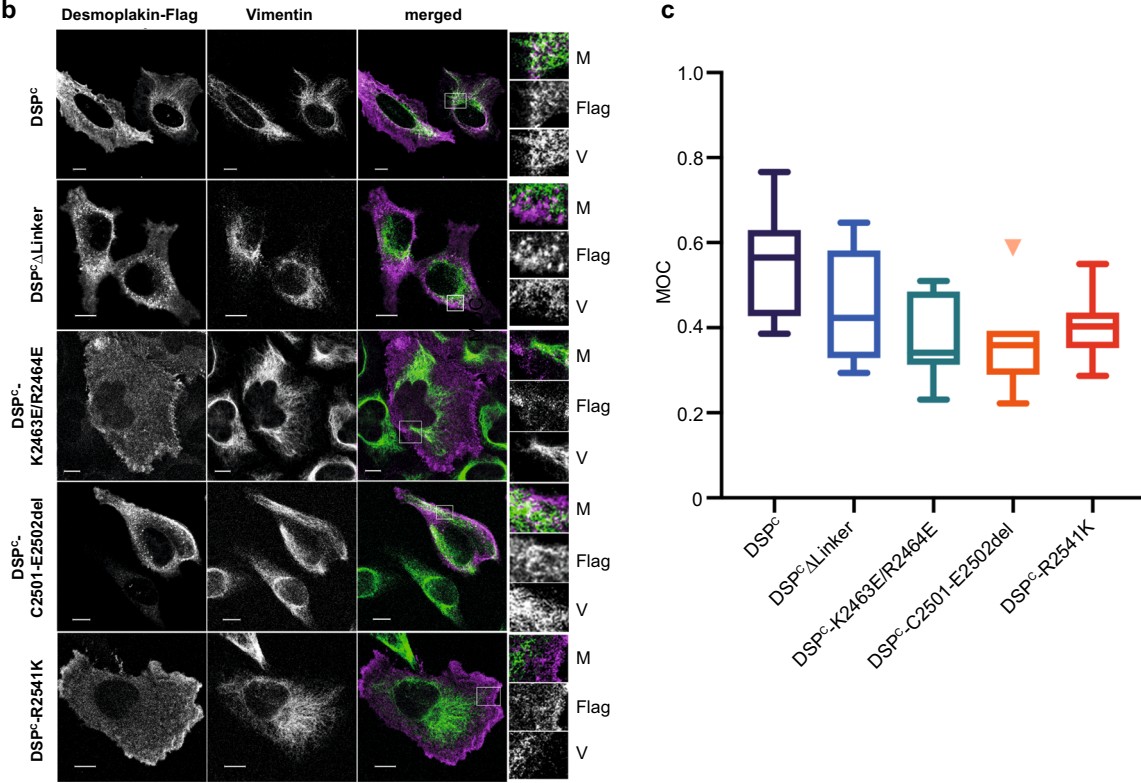

**Fig. 3 Targeting to vimentin intermediate filaments requires the desmoplakin linker and is compromised by mutations within its basic groove. a** Electrostatic surface potential and ribbon representation for the desmoplakin linker domain structural model derived from the I-TASSER server (C-score = 0.51). Ribbon representations highlight the position of basic residues K2463 and R2464 (stick format) in the groove (shown in black), residues C2501 and E2502 that are lost in ARVC mutation C2501-E2502del and residue R2541 which is mutated in ARVC (shown in red). **b** A desmoplakin DSP$^C$ construct encoding residues T1960–A2822 of human desmoplakin with a C-terminal FLAG tag was transfected into HeLa cells and the cells were stained with anti-FLAG and anti-vimentin antibodies. Desmoplakin is stained purple and vimentin is stained green in the merged images. The boxed areas are expanded on the far right-hand side. M merged image, FLAG desmoplakin staining, V vimentin staining. Bars, 10 μm. **c** Manders' overlap coefficient (MOC) was calculated for each image and is shown as a Tukey box plot with the median and 25th and 75th percentiles of each distribution. An unpaired *t* test with Welch's correction was performed on the data: DSP$^C$ versus DSP$^C$ΔLinker, not significant; DSP$^C$ versus DSP$^C$-K2463E/R2464E, $p = 0.01$; DSP$^C$ versus DSP$^C$-C2501-E2502del, $p = 0.007$; DSP$^C$ versus DSP$^C$-R2541K, $p = 0.01$. At least five fields of view were analysed for each experiment. z-stacks were taken for each field and overlap coefficients calculated for each individual z-stack. An average overlap coefficient was then calculated for each experiment and each experiment was repeated two to three times.

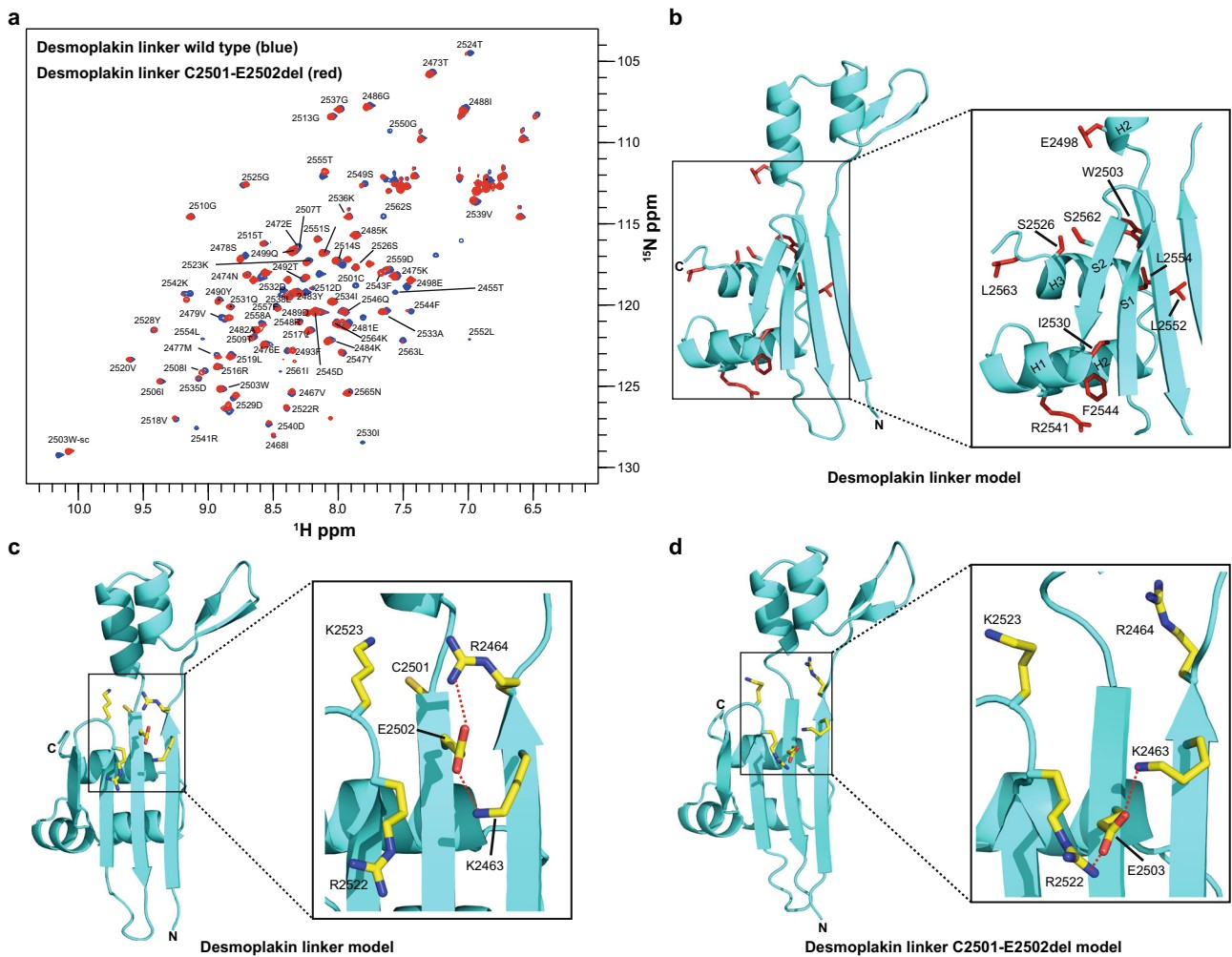

**Fig. 4 ARVC mutation C2501-E2502del causes only minor perturbations in the desmoplakin linker domain structure. a** Superimposed two dimensional $^1H,^{15}N$-resolved NMR spectra of wild-type (blue) and ARVC C2501-E2502del mutant (red) desmoplakin linker proteins. **b** Mapping chemical shift perturbations between the desmoplakin linker wild-type and C2501-E2502del onto the desmoplakin linker model structure. Residues with the largest chemical shift perturbations are shown (stick format and coloured red). **c** Ribbon diagram of the desmoplakin linker domain model showing the putative salt bridge interactions (red dashed lines) between E2502 and K2463/R2464. The box shows a close up view of the relevant interactions. **d** Ribbon representation of the desmoplakin linker domain C2501-E2502del model derived from the I-TASSER server (C-score = 0.19). The potential salt bridge interactions between E2503 and K2463/R2522 are shown (red dashed lines). The box shows a close up view of the relevant interactions.

perturbations were restricted to the Nt PR-like element (Supplementary Fig. 6b). It is possible that these changes adversely impact IF binding, explaining the lower MOC relative to wild-type desmoplakin (Fig. 3c). The desmoplakin linker model structure shows that R2541 protrudes from H2 of the Nt PR-like motif and mediates a salt bridge interaction with D2545 (Supplementary Fig. 6c). This ionic interaction most likely stabilises this helical region. In the case of the R2541K ARVC mutation, the ε-amino moiety of lysine is predicted to form a compensatory salt bridge interaction with the carboxylate group of D2545 (Supplementary Fig. 6d) which is likely to stabilise this helix, thereby preventing major structural rearrangements.

To further investigate the role of the desmoplakin linker in IF binding it (i.e. residues Q2454–N2565) was purified to homogeneity, as were mutants K2463E/R2464E, E2495K/C2497R and S2526K/Q2527K. All mutant desmoplakin linker proteins were folded, as indicated by the similarity of their $^1H,^{15}N$ resolved spectra to that of the wild-type protein (Supplementary Fig. 7a). Residues K2463, R2464, E2495 and C2497 are found in the basic groove (equivalent to periplakin residues R1655, R1656, K1687 and R1689, respectively), while residues S2526 and Q2527 are

beside the groove (Fig. 5a). The wild-type desmoplakin linker showed very weak binding to the vimentin$^{ROD}$ protein by MST, as did the K2463E/R2464E and S2526K/Q2527K mutant linker proteins (Fig. 5b, Supplementary Fig. 7b, Supplementary Table 2). Interestingly, the E2495K/C2497R mutant revealed enhanced binding to the vimentin$^{ROD}$ when compared to the wild-type desmoplakin linker protein (Fig. 5b), with an estimated $K_D$ of 600 ± 70 μM. Thus, increasing the basic character of the groove in the desmoplakin linker (Fig. 5c) led to significantly enhanced interactions with vimentin, although this was still considerably weaker than that of the periplakin linker protein ($K_D = 70.5 \pm 3.8$ μM). This is consistent with desmoplakin employing its linker domain and three PRDs to tether IFs, whereas periplakin relies on its linker domain alone.

Binding data indicate that the desmoplakin linker binds much less tightly to vimentin$^{ROD}$ than the corresponding region of periplakin. This finding was confirmed using NMR binding assay and full-length vimentin (vimentin$^{FL}$) protein (residues M1–E466). NMR experiments were carried out in the absence of salt to limit vimentin polymerisation into filaments that are too large to be suitable for characterisation of interactions. In the

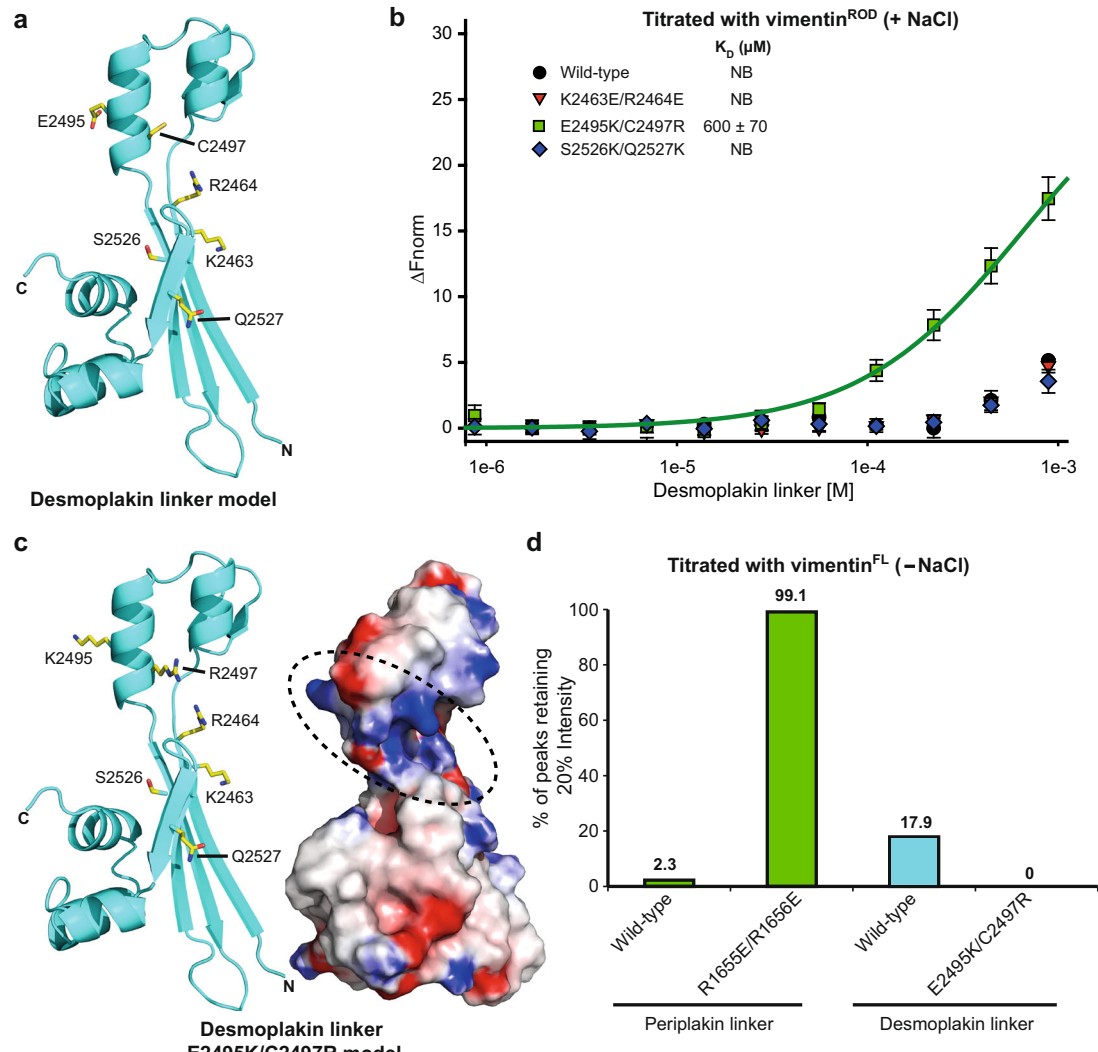

**Fig. 5 The desmoplakin linker domain shows very weak binding to the vimentin[ROD] but mutation E2495K/C2497R enhances this interaction. a** Ribbon representation for the putative desmoplakin linker domain structure highlighting the position of residues that were mutated (stick format) and assessed for binding to vimentin[ROD] by MST. **b** The binding of the desmoplakin linker and mutants to the vimentin[ROD] was examined by MST binding analysis. Increasing concentrations of the linker domain (780 nM to 800 μM) were incubated with 50 nM vimentin[ROD] and the MST results are plotted as $\Delta F_{norm}$ versus linker concentration. Data are shown as means ± SD of six independent experiments. Only the E2495K/C2497R mutant demonstrated significant binding to vimentin[ROD] and could be fit to a one-site ligand binding model to give an estimated $K_D = 600 ± 70$ μM. NB no binding. **c** Ribbon representation and electrostatic surface potential for the desmoplakin linker domain model encompassing the E2495K/C2497R double variant. A potential vimentin docking site is also highlighted (black dashed circle). **d** Histogram showing the percentage of [1]H, [15]N amide peaks retaining more than 20% of their peak intensity on addition of vimentin[FL] (200 μM) to periplakin and desmoplakin linker proteins (100 μM) in the absence of salt.

absence of salt full length vimentin forms functional tetramers[30], which were added to [15]N-labelled linkers from periplakin and desmoplakin to respective molar ratios of 0.1:1, 0.5:1, 1:1 and 2:1. Upon interaction with vimentin periplakin's linker displayed progressive [1]H,[15]N peak broadening, indicating slow exchange on the NMR timescale. The signals broadened dramatically, with only 2.3% of the linker amide peaks retaining at least 20% of their starting intensities at half equimolar ligand concentration (Fig. 5d, Supplementary Fig. 8). This suggests that the periplakin linker assembles on vimentin[FL] tetramers to form large, stable, slowly tumbling complexes. A similar, albeit less dramatic effect was observed when vimentin[FL] was added to the [15]N-labelled desmoplakin linker. In this case, 17.9% of peaks retained at least 20% of their starting intensity, indicating that the desmoplakin linker interaction with vimentin is weaker than that of the periplakin linker. Binding of the periplakin mutant R1655E/R1656E was compromised as expected when compared to the

wild-type periplakin linker with 99.1% of peaks retaining 20% intensity. Similarly, binding of the desmoplakin mutant E2495K/C2497R was enhanced when compared to that of the wild-type desmoplakin linker, with no peaks retaining 20% of their starting peak intensity (Fig. 5d). Thus the NMR binding results mirror those by MST and cellular co-localisation, consistent with electrostatic forces within the groove driving vimentin recognition.

**A model for the periplakin linker–vimentin complex.** Vimentin is the best understood IF, and multiple structures are available to build models of its assemblies. Monomeric vimentin is a rod-shaped protein consisting of an α-helical central region that is flanked by non-helical head and tail domains (Supplementary Fig. 1b). Vimentin monomers have a strong tendency to dimerise via the formation of α-helical coiled coil dimers. Dimers then associate in half staggered anti-parallel fashion to form tetramers

that laterally associate to form octamers and higher order oligomers[31]. The vimentin dimer serves as the elementary building block for IF assembly, and displays multiple acidic patches on its surface that could be recognised by basic residues in the linker domain groove. Periplakin linker domain-vimentin complexes were modelled using the high ambiguity driven protein–protein DOCKing (HADDOCK) programme[32]. The periplakin residues identified as being crucial for co-localisation with vimentin in transfection experiments (i.e. R1655, R1656, K1687, R1689, R1713 and K1714) were used to restrain docking to conserved negatively charged residues within available vimentin structures (Supplementary Fig. 9, Supplementary Table 3). In the resulting models vimentin consistently slotted into the periplakin linker positive basic groove with minimal structural rearrangement. This was not unexpected given the breadth of the groove and the dimensions of the vimentin dimer, which consists almost entirely of an α-helical coiled coil with multiple acidic patches along its length. The angle of vimentin ingress and egress varied, and several of vimentin's acidic patches mediated favourable interactions. The two lowest energy complex models obtained consisted of the periplakin linker domain interacting with a vimentin fragment encompassing residues T99–L189 (PDB 3S4R) and E153–H238 (PDB 3SWK) (Fig. 6a, b, Supplementary Table 3). In complex model 1 (Fig. 6a) electrostatic interactions were observed between vimentin residues D162 and D166 and the periplakin linker groove side chains R1689 and R1713. In complex model 2 the periplakin linker–vimentin interface was stabilised by ion pair interactions mediated by vimentin residues E172, D176 and E180 and several basic side chains of the periplakin linker groove (Fig. 6b). In addition, the vimentin residue E187 was in close proximity to R1689 of periplakin underlying an additional potential electrostatic interaction. To validate these electrostatic docking modes a series of charge reversal mutations were designed in the vimentin[ROD] fragment and tested for effects on linker recognition (Fig. 6c, Supplementary Fig. 10, Supplementary Table 4). Residues D162, E172, D176, E180, E187 and E229 are situated in acidic helical patches and were mutated to lysines. Proton NMR spectra of the vimentin mutants (Supplementary Fig. 11) demonstrate that these protein are correctly folded. Binding interactions of these mutants with wild-type periplakin linker was measured by MST. Two of the substitutions, D176K and E187K, totally abolished the interaction of the vimentin[ROD] with the periplakin linker, suggesting that they contribute to a docking site. Two mutants, E172K and E229K exhibited a moderate increase in linker binding affinity whilst one, E180K, exhibited a larger increase in affinity. One possible explanation for the latter is that the lysine residue can form an ionic interaction with E1692 which borders the basic groove of the periplakin module. The D162K mutant displayed wild-type binding characteristics suggesting that this residue is not involved in linker recognition. Overall, the data demonstrate the importance of residues D176 and E187, and make complex model 2 the more likely candidate for periplakin linker–vimentin binding. In this model vimentin residues E172 and D176 from coil 1B are recognised by periplakin R1655 and R1713, while vimentin's E180 contacts periplakin residues R1689 and K1714 (Fig. 6b). The importance of periplakin residue R1713 and vimentin residue D176 was confirmed in experiments showing that binding of periplakin mutant R1713E to wild-type vimentin[ROD] was reduced whereas its binding to vimentin mutant D176K was enhanced (Fig. 6d, Supplementary Fig. 12). Collectively, the presence of residues E172, D176, E180 and E187 on a continuous acidic surface that is conserved in IFs (Supplementary Fig. 6) suggests that electrostatic interactions between basic residues in linker domain grooves and acidic IF residues may be a widely used mechanism of cytoskeletal attachment.

## Discussion

Linker domains play important and diverse roles in plakin biology that can be attributed to their universal and critical IF-tethering function. They are found in five plakin proteins, each of which has a unique and important role in the development and maintenance of tissues that undergo mechanical stress. The periplakin linker forms an elongated bilobed domain that frames an electropositive groove that represents the functional epicentre of the domain (Fig. 1). The three dimensional structures of the coiled-coil rod domains of vimentin and keratin IFs have been determined and these reveal multiple acidic patches along their cylindrical surfaces[33–35]. Studies of mutations in the vimentin[ROD] fragment and the periplakin linker module indicate that the linker domain accommodates cylindrical IF ligands through electrostatic interactions. This mechanism is reminiscent of the mode by which PRDs interact with IFs[3]. While PRDs are larger than linker domains encompassing 4.5 PR motifs rather than the pair of PR-like sequences found in linkers, they also offer a distinct positively charged groove. Nevertheless the linker's IF recognition mechanism resembles that of the PRD groove which accommodates cylindrical IF ligands through electrostatic attraction. Charge reversal substitutions in the periplakin and desmoplakin linkers compromise their targeting and co-localisation with vimentin IFs (Figs. 1 and 3), mirroring the effects of mutations in the envoplakin PRD groove that compromise targeting and co-localisation of its assembly with vimentin[3]. Similarly charge reversal mutations in the vimentin[ROD] abolish periplakin linker binding in a comparable way to how they abrogate envoplakin PRD binding to vimentin[3]. Hence a holistic mechanism is emerging in which proximal linker and PRD domains both employ electrostatic attraction mediated by their respective basic grooves to provide the avidity needed for stable IF tethering. Our experiments show that vimentin residues D176 and E187 which emanate from coil 1B are vital for the interaction with the periplakin linker module (Fig. 6). There may be some variation in residues required for binding other IF proteins as residue D176 is conserved in desmin and keratins but E187 is not (Supplementary Fig. 2). In previous work we demonstrated the importance of vimentin residues D112 and D119, protruding from coil 1A, for binding to envoplakin PRD[3]. Hence, there is a distinct possibility that the binding of the periplakin linker and envoplakin PRD to vimentin is not mutually exclusive and the contribution of both may be required for strong attachment of the periplakin–envoplakin heterodimer to the IF cytoskeleton.

Interestingly, the periplakin linker appears to show stronger binding to vimentin than does the desmoplakin linker (Figs. 2a and 5b), although it is not as strong as that of the envoplakin PRD ($K_D = 19.1$ μM)[3]. Both linkers contain similar numbers of positively charged residues in their groove areas (Figs. 1 and 3) so this is not simply a matter of basic character and forces other than electrostatic interactions, including steric fit and hydrophobic interactions, may also be in play. Increasing the basic character of the desmoplakin linker groove does enhance its affinity for the vimentin[ROD], although not to the level of the wild-type periplakin linker (Fig. 2a and 5b), indicating that charge is important but insufficient for tight interactions. We speculate that evolutionary pressure on periplakin has led to the development of high binding affinity of its linker for vimentin, enabling it to bind IFs in tissues where its heterodimerisation partner envoplakin is not expressed. Loss of this affinity would render periplakin entirely dependent upon heterodimerisation with envoplakin for IF binding. By contrast evolutionary pressure to retain desmoplakin linker binding may not be as strong as the desmoplakin tail encompasses three PRDs, each of which is capable of binding IFs. In desmoplakin the role of the linker may be to provide proper geometric positioning of the two flanking PRDs. Delineation of

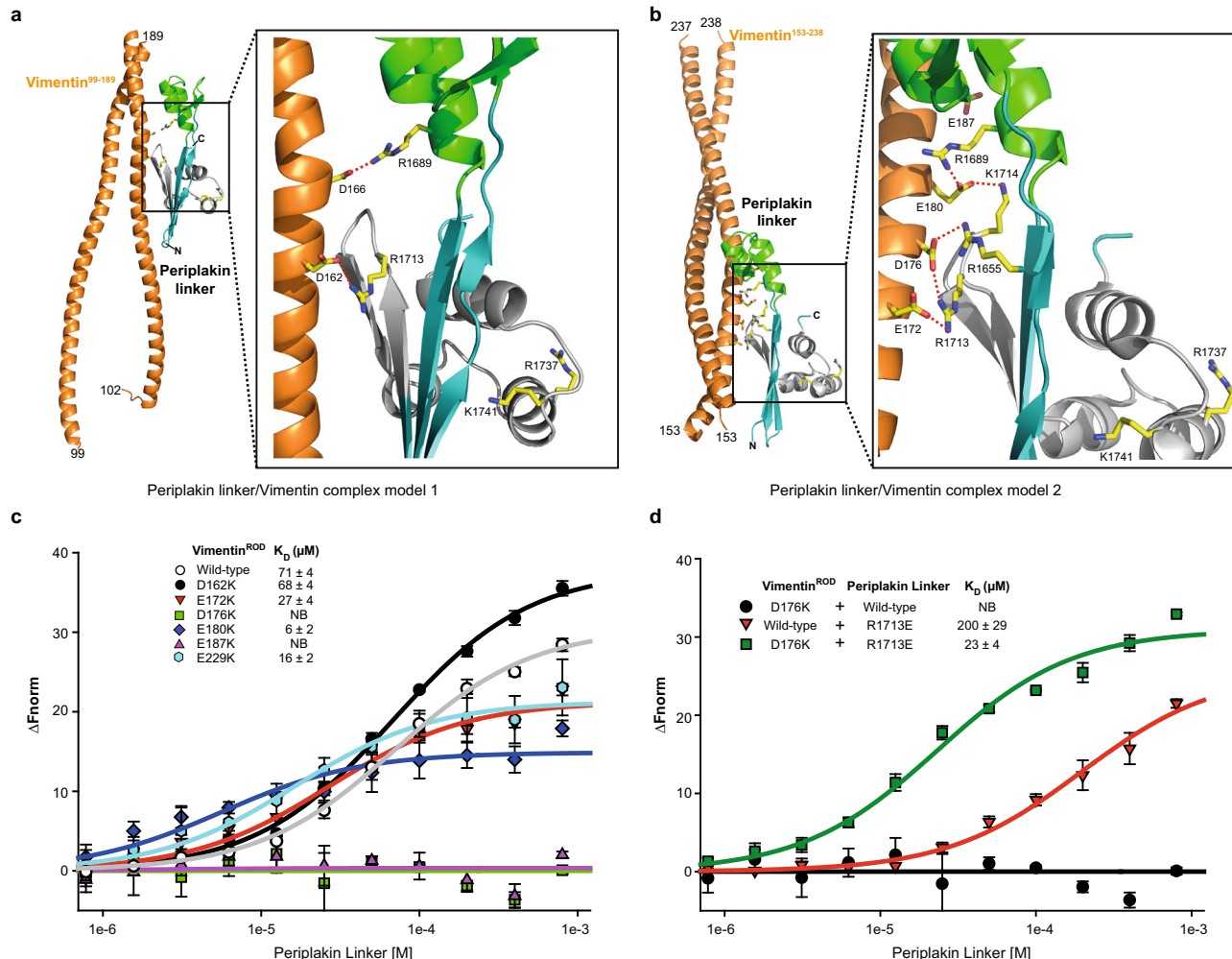

**Fig. 6 Modelling and binding studies of the periplakin linker domain-vimentin complex. a, b** The two lowest energy periplakin linker/vimentin complex models generated by HADDOCK. Model 1 corresponds to the I-TASSER derived periplakin linker structure docked with vimentin$^{99-189}$ (PDB entry 3S4R). Model 2 is comprised of the periplakin linker structure docked to vimentin$^{153-238}$ (PDB entry 3SWK). The putative interfaces are stabilised by salt bridge interactions (red dashed lines). **c** Binding of the wild-type periplakin linker to mutated vimentin$^{ROD}$ proteins measured by MST. Increasing concentrations of the wild-type periplakin linker domain (0.78 μM to 800 μM) were incubated with 50 nM labelled mutant vimentin$^{ROD}$ protein. The baseline normalised fluorescence $\Delta F_{norm}$ is plotted against increasing concentration of the linker. Data points are mean ± SD from four independent experiments and were fit to a one-site binding model. The binding affinity $K_D$ was estimated from these fits to be D162K = 68 ± 4, E172K = 27 ± 4, E180K = 6 ± 2 and E229K = 16 ± 2 μM. D176K and E187K did not demonstrate any interactions with the periplakin linker (NB no binding). The binding curve for the wild-type vimentin$^{ROD}$ protein with periplakin linker ($K_D$ 71 ± 4 μM) from Fig. 2 is added for comparison. **d** Binding of periplakin linker mutant R1713E to wild-type and D176K vimentin$^{ROD}$ proteins measured by MST. Increasing concentrations of the periplakin linker domain (800 to 0.78 μM) were incubated with 50 nM labelled vimentin$^{ROD}$ proteins. The baseline normalised fluorescence $\Delta F_{norm}$ is plotted against increasing concentration of the linker. Data points are mean ± SD from three to four independent experiments and were fit to a one-site binding model. Periplakin mutant R1713E binding affinity $K_D$ was estimated from these fits to be 200 ± 29 (wild-type vimentin$^{ROD}$) and 23 ± 4 μM (D176K mutant vimentin$^{ROD}$). Wild-type periplakin linker showed no binding to D176K vimentin$^{ROD}$ (this data from Fig. 6c is added for comparison).

multivalent binding modes requires further analysis, but could involve sliding of binding grooves along filaments to secure adaptive attachments.

The clinical effects of desmosomal protein mutations can now be interpreted in light of the linker mechanism. Deletion of two residues within the desmoplakin linker domain (C2501 and E2502) results in ARVC[28]. These non-positively charged residues protrude from the central groove and are located in an equivalent position to the periplakin linker region 1694 DWEEI 1699, which is critical for IF targeting[22]. Loss of desmoplakin residues C2501 and E2502 result in subtle rearrangements in the positively charged groove (Fig. 4d) and partial co-localisation with vimentin (Fig. 3b). Thus it appears that even minor changes in the groove affect IF co-localisation, albeit not to a dramatic extent. We recognise that we are

measuring co-localisation with vimentin IFs in our experiments, whereas in cardiomyocytes desmoplakin interacts with desmin IFs. However, given the high degree of similarity between these two IF proteins it is likely that the mechanism by which the desmoplakin linker domain binds desmin IFs is similar to that by which it engages vimentin IFs, i.e. via electrostatic interactions between positively charged residues in the linker domain groove and negatively charged side chains on IF rods.

In summary, our results provide a mechanistic basis for understanding of plakin protein linker domain-IF interactions. Linker domains interact with IFs via electrostatic interactions with IF rods slotting into electropositive grooves. The role of plakin proteins in cell-cell and cell-matrix adhesion, and other cell biological processes such as cell migration, can now be more

precisely probed and the effects of linker domain mutations in disease can be rationalised. For example, sequencing of malignant melanomas has identified 5 somatic mutations in desmoplakin linker residue R2465 (R2465K)[36]. This N-terminal residue is highly conserved among plakin family members (except periplakin) and is predicted to stabilise the PR2-like motif by forming a hydrogen bonding interaction with the carboxyl group of Q2499 which emanates from H2. Our modelling suggests this interaction would not be preserved in the melanoma-linked R2465K mutant, potentially resulting in a loss of linker domain stability and IF binding. Similarly, periplakin cancer-linked mutations including R1655W, R1656C, R1713M and R1737H may alter the electrostatic IF binding function of its linker domain[36–38]. From a structural biology perspective the aim will now be to produce predictive models and structures of the multivalent complexes between periplakin/envoplakin heterodimers and desmoplakin homodimers, and IF proteins.

## Methods

**Transfection, immunofluorescence microscopy and westerns.** DNA encoding the following human protein sequences were subcloned into expression vector pcDNA3.1(−) (Life Technologies): periplakin residues M1588–K1756 with a C-terminal HA tag (YPYDVPDYA) and desmoplakin residues T1960–A2822 with a C-terminal Flag tag (DYKDDDDK) (Supplementary Fig. 3a) (Supplementary Table 5). Mutant periplakin and DSPs were produced using the QuikChange Lightening site directed mutagenesis kit (Agilent Technologies). Mycoplasma-free HeLa cells were obtained from Cancer Research UK (London Research Institute) and routinely cultured in Dulbecco's modified Eagle's medium supplemented with 10% foetal calf serum, penicillin (50 U/ml) and streptomycin (50 μg/ml). Cells were passaged by treatment with trypsin/EDTA. Constructs were transfected into cultured cells using GeneJammer transfection reagent (Agilent Technologies) according to the manufacturer's protocol. For immunofluorescence microscopy cells were grown on glass coverslips in complete media for 24–36 h prior to transfection. At 48 h following transfection cells were fixed for 10 min in 4% paraformaldehyde and permeabilised for 2 min with 0.1% Triton X-100. Cells were co-stained with either anti-HA (Cell Signalling, catalogue number sc-7392, 1000-fold dilution) or anti-Flag (Sigma-Aldrich, F1804, 1000-fold) and anti-vimentin (Cell Signalling, 3932, 50-fold) antibodies, followed by the appropriate AlexaFluor-conjugated secondary antibodies (Invitrogen, A-11019 and A-11070, 1000-fold). Coverslips were mounted onto microscope slides using SlowFade Gold antifade reagent (Life Technologies). Images were taken using a Zeiss LSM510 META confocal system with ×63 oil immersion objective (NA 1.4). Co-localisation of plakin constructs and vimentin was quantified using the JACoP plugin from ImageJ (Rasband, WS, ImageJ, National Institutes of Health, USA; https://imagej.nih.gov/ij). A set of commonly used co-localisation indicators was examined by visual inspection of the staining using the decision tree proposed by the JACoP developers[39]. Manders' coefficient was chosen as the most appropriate method because it measures the fraction of pixels with positive values in two channels regardless of signal levels. This is important because the expression of transiently transfected proteins, and hence the signal in one channel may vary between images. For western blotting transfected cells were lysed in sodium dodecyl sulfate (SDS) sample buffer, resolved by SDS-polyacrylamide gel electrophoresis and transferred to Hybond-LFP polyvinylidene difluoride membrane. Blots were probed with anti-HA (Cell Signalling, sc-805, 500-fold), anti-Flag (Sigma-Aldrich, F7425, 3000-fold) and anti-actin (Sigma-Aldrich, A5441, 20,000-fold) antibodies, followed by the appropriate HRP-conjugated secondary antibodies (Dako, P0448 and P0447, 1000-fold).

**Purification of periplakin and desmoplakin linker domains.** DNA encoding the linker domains of human periplakin (residues K1646–K1756) or desmoplakin (residues Q2454–N2565) were cloned in-frame with glutathione S-transferase (GST) in expression vector pGEX-6P-1 (GE Healthcare) (Supplementary Table 5). Linker domain mutants were produced using the QuikChange Lightening site-directed mutagenesis kit (Agilent). Constructs were expressed in *E. coli* strain BL21 (DE3). Bacterial cultures were grown in LB media or minimal medium supplemented with $^{15}NH_4Cl$ for NMR studies. Cultures were grown at 37 °C until the absorbance at 600 nm had reached 0.6, the temperature was then lowered to 18 °C, expression was induced with 1 mM isopropyl-β-D-thiogalactopyranoside and the cultures were grown for a further 18 h. Cells were harvested by centrifugation and resuspended in 100 mM NaCl, 20 mM sodium phosphate (pH 7.4) with protease inhibitors (Roche). Cells were lysed with an Emulsiflex system (Avestin) and the lysates cleared by centrifugation and filtered. GST-fused proteins were purified by glutathione affinity chromatography. Briefly, cell lysates were loaded onto 5 ml GSTrap HP columns (GE Healthcare), columns were washed with 150 mM NaCl, 20 mM sodium phosphate (pH 7.4) and fusion proteins eluted with 30 mM glutathione, 250 mM NaCl, 200 mM Tris-Cl (pH 8.0). Fusion proteins were then

incubated overnight at 4 °C with PreScission protease (GE Healthcare), the cleaved GST removed by binding to GSTrap columns and linker proteins further purified by size exclusion chromatography using Superdex S75 columns pre-equilibrated with 100 mM NaCl, 20 mM sodium phosphate (pH 7.2) for NMR samples or 150 mM NaCl, 20 mM HEPES (pH 7.5) for binding studies. Proteins were kept at 4 °C or glycerol was added to 20% and the proteins stored at −80 °C.

**Purification of vimentin^ROD and vimentin^FL proteins.** Human vimentin^ROD (residues T99–I249 with a non-cleavable His tag) and full-length vimentin (residues M1–E466) proteins were expressed in bacteria and purified as described[3]. Vimentin^ROD mutants were produced using the QuikChange Lightening site-directed mutagenesis kit (Agilent). Vimentin^ROD proteins were examined by proton NMR to ensure that the proteins were properly folded and similar in structure (Supplementary Fig. 11). Proteins were exchanged into 20 mM phosphate buffer, pH 7.0 containing 10% $D_2O$ and 0.02 mM 4,4-dimethyl-4-silapentane-1-sulfonic acid (DSS) as an internal chemical shift reference. Protein concentration was adjusted to 200 or 500 μM and 200 μl samples were transferred to 3 mm NMR tubes. The NMR spectra for the protein and mutants were collected at 25 °C using a Varian Unity INOVA 600-MHz spectrometer. All spectra were collected with 64 steady-state scans, an acquisition time of 2 s, a 90° proton pulse of ~12.2 μs, and the number of acquired scans was 384 per free induction decay. The data were apodized with an exponential window function corresponding to a line broadening of 0.3 Hz, Fourier-transformed, phased and baseline-corrected for comparison.

**MST analysis of linker–vimentin binding.** Purified vimentin^ROD protein was labelled using the Monolith NT His-Tag Labelling Kit RED-tris-NTA (Nano-Temper Technologies) to produce 100 nM NT647 fluorescent dye-labelled target in 150 mM NaCl, 20 mM HEPES (pH 7.5) with 0.015 % Tween 20. Linker proteins were exchanged into the same buffer using PD MiniTrap G-25 gravity columns (GE Healthcare) and concentrated to generate a series of twofold dilutions with concentrations ranging from 1.6 mM to 1.56 μM. Each ligand dilution was mixed with an equal volume of labelled vimentin^ROD leading to a final concentration of 50 nM vimentin^ROD and final linker concentrations ranging from 800 μM to 780 nM. A maximum concentration of 800 μM linker protein was used to prevent non-specific interactions. After incubation for 10 min at room temperature, the samples were loaded into standard capillaries (NanoTemper Technologies) and MST data was collected at 25 °C, 40% LED power and medium MST power. No sign of adsorption or aggregation were found in any of the data traces. To test the effect of salt on linker protein-vimentin^ROD interactions binding experiments were performed in 150 mM NaCl (as above), 50 mM NaCl and 10 mM NaCl.

**NMR analysis of linker–vimentin binding.** All samples contained 100 μM $^{15}N$-labelled wild-type and mutant periplakin and desmoplakin linker proteins in 20 mM Tris-Cl, (pH 7.0) and 1 mM DTT. Heteronuclear single quantum coherence spectra were recorded at 298 K on a Varian INOVA 600 MHz spectrometer equipped with a triple resonance cryogenically cooled probe. Binding to vimentin was measured by following the changes in peak intensity of the $^{15}N$-labelled linker proteins upon addition of unlabelled vimentin^FL to final concentrations of 10, 50, 100 and 200 μM.

**NMR assignment of the desmoplakin linker.** Samples of $^{15}N$, $^{13}C$-labelled desmoplakin linker proteins (100–500 μM) were prepared in 20 mM Hepes (pH 7.5), 50 mM NaCl, 10% $D_2O$. All experiments were performed on a 600 MHz Varian Inova spectrometer equipped with a cryogenically cooled triple resonance probe at 298 K. Backbone resonances were assigned using BEST versions of the HNCO, HNCACO, HNCA, HNCOCA, HNCACB and HNCOCACB[40] using an interscan delay of 300 ms. All experiments were acquired collecting 420 data points in the direct dimension with a sweep width of 11.2 ppm and 32 increments in the $^{15}N$ dimension with a sweep width of 30 ppm. The HNCO and HNCACO experiments were acquired using 8 scans and 48 increments in the CO dimension with a sweep width of 16 ppm. The HNCA and HNCOCA were acquired using 16 scans and 64 increments in the CA dimension with a sweep width of 30 ppm. The HNCACB and HNCOCACB were acquired using 16 scans and 64 increments in the CA dimension with a sweep width of 80 ppm. Spectra were processed using NMRPipe[41] and analysis was performed using Sparky (Goddard TD and Kneller DG, SPARKY 3, University of California, San Francisco) and CCPN Analysis[42].

**Far ultraviolet circular dichroism spectroscopy.** CD spectra were measured on a Chirascan CD spectrometer (Applied Photophysics) using a 1 cm path length cuvette and a scanned wavelength range of 200–250 nm with sampling points every 1 nm. Data were processed using an Applied Photophysics Chirascan viewer and Microsoft Excel.

**Modelling the periplakin linker–vimentin complex.** The interaction between the periplakin linker domain and vimentin was modelled with HADDOCK[32]. Periplakin residues were classified as active in vimentin binding based upon the results of co-localisation and binding experiments. 'Passively involved' residues were selected automatically. To generate representative structural models molecular

docking experiments were carried out with available vimentin fragment structures encompassing the entire central rod domain of vimentin. These included PDB entries 3G1E (residues N102–L138), 3S4R (T99–L189), 3SWK (E153–H238), 3UF1 (L146–I249) and 3KLT (D264–K334). Vimentin residues contacted by the linker were predicted from conservation of sequence motifs, negative charge and surface exposure. Vimentin residues selected for use as ambiguous interaction restraints to drive the docking process are listed in Supplementary Table 3.

**Structural modelling of linker domains.** The structures of periplakin linker (residues K1646–K1756), desmoplakin linker (residues Q2454–N2565), desmoplakin linker C2501-E2502del (residues Q2454–N2565 with C2501 and E2502 omitted) and plectin linker (residues K4266–A4377) domains were generated using the I-TASSER (Iterative Threading ASSEmbly Refinement) server[43]. Briefly, the target sequences were initially threaded through the Protein Data Bank (PDB) library by the meta threading server, LOMETS2. Continuous fragments were excised from LOMETS2 alignments and structurally reassembled by replica-exchange Monte Carlo simulations. The simulation trajectories were then clustered and used as the preliminary state for second round I-TASSER assembly simulations. Finally, lowest energy structural models were selected and refined by fragment-guided molecular dynamic simulations to optimise hydrogen-bonding interactions and remove steric clashes. Models were ranked based on their I-TASSER confidence (C) score (range −5 to +2 with a higher score correlating with an improved model).

**Statistics and reproducibility.** For quantification of immunofluorescent microscopy at least five fields were examined for each experiment, with each field containing two to four transfected cells. z-stacks (slice thickness 0.7 μm) were taken for each field and overlap coefficients calculated for each individual z-stack. An average overlap coefficient was then calculated for each experiment and each experiment was repeated two to three times. Unpaired t tests with Welch's correction was performed on the data. For MST binding studies data from three to six independent experiments were analysed (MO.Affinity Analysis, NanoTemper Technologies) and the results plotted and fit to a one-ligand binding model with SigmaPlot (Systat Software).

**Reporting summary.** Further information on research design is available in the Nature Research Reporting Summary linked to this article.

## Data availability
The data that support the findings of this study are either available within the paper (and its Supplementary information files) or are available from the corresponding author upon reasonable request.

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

## Acknowledgements
We thank Graeden Winkelaar and Yilin Wang for discussions and assistance, and the University of Birmingham Protein Expression Facility and Henry Wellcome Building for Biomolecular NMR Spectroscopy (HWB-NMR) for access. MO received funding for this work from the Campus Alberta Innovates Programme (RCP-12-002C), Alberta Prion Research Institute/Alberta Innovates Bio Solutions (201600018), NSERC RGPIN-2018-04994 and TMIC grants from Genome Canada and the Canada Foundation for Innovation Major Sciences Initiative fund and Wellcome Trust Research Project 094303/Z/10/Z. F.M. is funded by Wellcome Trust grant 099266/Z/12/Z. T.K. is funded by BBSRC grant BB/P009840/1.

## Author contributions
E.O. performed all the microscopy and immunofluorescence studies. F.M. produced all the structural models and C.T. carried out MST-binding assays. P.R-Z., C.A-J., T-H.H., C.F., P.S. and J.K. purified proteins and produced NMR spectra, which were analysed by T.K. and M.J. P.R-Z. and C.F. carried out NMR-binding assays. M.C and C.T. produced constructs and M.C. and M.O. designed the experiments. F.M., E.O., C.T., M.C. and M.O. analysed the data and wrote the paper.

## Competing interests
The authors declare no competing interests.
