## [Peer Review File · Communications Biology]

Reviewers' comments:

Reviewer #1 (Remarks to the Author):

Odintsova et al. report the biophysical and functional characterization of the plakin linker domain. Unlike other plakin proteins, such as envoplakin and desmoplakin, periplakin does not contain plakin repeat domain (PRD) but has the conserved linker module. Previous studies showed that PRD interacts with intermediate filament (IF) protein mostly through its positively charged groove, but the molecular mechanism of the interaction between the linker and IF proteins has not been revealed yet.

Based on the previously published structure of periplakin linker, the authors propose that positively charged groove of the periplakin linker would be involved in the interaction with vimentin. To prove this proposal, the authors design charge reversal mutants on the putative IF binding groove of the periplakin linker, and perform *in vitro* binding assays and immunofluorescence microscopy using these mutants, showing that the positively charged residues are important for the interaction with vimentin *in vitro* and co-localization with vimentin IF *in vivo*. The authors then apply similar methods to desmoplakin linker domain, to demonstrate that positively charged residues on the groove in the linker domain contribute on cellular co-localization of vimentin IF and the interaction with vimentin *in vitro*, although its binding affinity is weaker than that of the periplakin linker. Finally, the authors propose a structural model of periplakin linker-vimentin complex supported by binding affinity measurements with charge reversal mutants of vimentin rod domain. These results will provide interesting insights into how the plakin linker recognizes IF proteins and will be the basis of future studies on the mechanism of plakin dimer-IF interaction.

In the opinion of this reviewer, the authors should address and clarify the following issues and tone down those parts of their conclusions that are not fully supported by the experiments in current manuscript.

Major comments:

1. Analysis of the previously released crystal structure of periplakin linker (PDB ID 4Q28) is an important starting point to address that periplakin linker forms a positively charged groove where vimentin IF binds. However, this crystal structure forms a stable beta-sheet mediated by a hexa-histidine tag sequence and crystal contacts with neighboring molecules in the crystal lattice, which are not physiologically relevant. It is not clear whether periplakin linker still forms a bi-lobed module connected by long beta-strands as shown in Figure 1a after removal of hexa-histidine residues from the structure and isolation of single protomer from the crystal lattice. The authors also mention that this structure has crystallization artifacts but the crystal structure was used to model the linker domain without experimental verification. This structure is very critical for this manuscript because all model structures including linker-vimentin complex were generated using this structure as a template. Therefore, it is necessary to clarify that Figure 1a represents the structure of periplakin linker in solution. Do you have any experimental evidence that periplakin linker forms a central beta-sheet as shown in Figure 1a, such as CD spectrometric analysis? Or, do you have any modeling data (of MD simulation) in the absence of hexa-histidine residues and crystal contacts?

More importantly, Figure 1a and the secondary structure presentation in Figure S2(a) do not match each other. Based on Figure S2 (and PDB), R1655 is located at the N-terminal region of the first beta-strand but it is shown near the C-terminal region of the first beta-strand in Figure 1a. Could you explain this? What residues form the first beta-strand in Figure 1a? If hexa-histidine residues are included in the structure representation (in Figure 1a), that is very misleading. Only the periplakin linker residues should

be presented. Also, please label the beta strands such as S1, S2, etc. in Figure 1a and Figure S2(a).

2. Related to point #1, computational modeling constitutes an important part of this manuscript. The accuracy and reliability of model is critical for the follow-up experiments and conclusions. In plectin linker and desmoplakin linker models (Figures 2d and 3a, respectively), structural variations of PR2-like domain and Nt PR-like domain are observed; for example, two beta-strands are missing in PR2-like domains. In contrast, the central beta-strands are well maintained in all plakin linkers. Do you think crystallographic artifacts in the template structure are removed during modeling? Do these models match well with secondary structure prediction analysis?

Also, model coordinate files in the supplementary are essential. It is hard to compare linker structures in the figures.

3. It is more appropriate to move the first section of results (lines 118 to 138, “Structure of the periplakin linker reveals a distinctive positively charged groove”) to the introduction or to combine it with the second section of results because not enough original data is included in this section. Especially, structural analysis of periplakin linker (lines 121 to 125 and 128 to 132) was already published by Kang et al., (2016), although they questioned the formation of a central beta-sheet structure in physiological conditions. In lines 132-133, the authors say the conserved N-terminal residues that could normally form part of the structure but it is not clear which residues are mentioned here. And, do the authors mean that these residues are conserved in all plakin proteins? Please clarify this.

4. The authors show that R2541K mutation of desmoplakin exhibited a reduced MOC, similar to K2463E/R2464E and explain this mutation at the molecular level using the desmoplakin linker model structure (lines 271-277). However, this structural analysis doesn't explain how R2541K mutation showed a similar effect to the double mutations in the basic groove (K2463E/R2464E). The authors only say that R2541K mutation still maintains the overall structural integrity, allowing partial co-localization with IFs (lines 274-277). Can the authors explain why R2541K shows a reduced MOC?

5. The authors present the model structure of the periplakin linker-vimentin complex by using the HADDOCK program and validation with affinity measurements of charge reversal mutants. However, the experimental validation does not fully support the complex model, because MST data using the mutants only show that D176, E180, and E187 are important for the interaction with the periplakin linker, but do not specify the interacting residues of the periplakin linker. If the authors can show that the abolished affinity of D176K mutant of vimentin(ROD) is recovered by using R1713D(or E) mutant of the periplakin linker, that can support the model.

Minor comments:

6. In Fig. 1b, Fig. 3b, please label each color.
7. In Fig. 6d, please include wild type data.
8. In Fig. S4 (c) and (d), it should be D2545, rather than D2541.
9. In line 201, it should be “1694-1698”, rather than “1694-1699”.
10. In line 202, reference in parenthesis should be shown as number (22)

Reviewer #2 (Remarks to the Author):

In this paper Odintsova and collaborators provide insights into how the plakin proteins, periplakin and desmoplakin, bind to its intermediate filament partner vimentin. In this study the team apply structure-based mutagenesis in order to identify key residues for the interaction of these proteins. The mutants selection is performed based on the structure of the periplakin linker domain, which is available in the PDB database with code 4Q28 (DOI: 10.2210/pdb4Q28/pdb). For desmoplakin, they use computational methods for the structural modelling of the linker domain. Reverse-charge mutations are introduced in residues located in an electropositive groove, which is proposed to be key in binding to acidic groups of vimentin via complementary electrostatic interactions. The effect of the mutants is tested in a cell culture model and further confirmed by microscale thermophoresis (MST). The authors apply computational simulations to elucidate different binding models of periplakin to vimentin by docking, providing specific electrostatic interactions between them.

This article is of interest, well-written, and the experiments seem to be well-conducted, however I believe there are some issues that need to be addressed.

Specific comments:

- 1) Figures 1b and 3b. The quantification of the co-localization experiments should be readdressed. Please, indicate the specific number of images per condition that were used to calculate the Manders' overlap coefficient and the number of independent experiments (or replicates) in the figure legends. A Tukey box plot showing the median and 25th and 75th percentiles for each distribution as well as the indication of the p-values is recommended. What are representing the error bars?
- 2) I have some concerns about the poor fit of the MST data for the mutations R1737E/K1741E in periplakin or the mutation E180K in vimentin (Figure 2a and Figure 6d). Could authors explain these behaviours? In addition, it may be helpful to have a supplementary table that includes the Kd values and r^2 for each independent experiment.
- 3) The authors claim that an electrostatic component is key for the interaction between periplakin and vimentin. This could be further supported with MST experiments at different salt concentrations. Could the salt concentration of the reaction have any effect on the Kds?
- 4) The PDB doi link (<http://dx.doi.org/10.2210/pdb4Q28/pdb>) for the structure with PDB code 4Q28 should be mentioned and cited.
- 5) In Supplementary Figure 1a, the addition of the domain organisation for BPAG1e and plectin would help the readers to better understand the manuscript.
- 6) Periplakin-vimentin complex computational modelling. In model 3 (the most supported model) the periplakin mutations R1689, R1655 and R1713 were identified as key-residues for the interaction. In my opinion, every single mutant should also be tested in order to confirm the inhibitory effect and validate the complex model. Moreover, these results, together with those for vimentin D176 and E187, should be mentioned in the abstract, since they contribute to pinpoint the impact of the manuscript. Since these two last single mutations completely abolish the interaction, a possible structural impact of these variants should also be analysed. Are these variants well folded when a lysine is introduced? Circular dichroism, SAXS, etc. could be useful to answer this question.

7) Need to discuss why vimentin E180K mutant stabilises the interaction with periplakin. Have authors tried to crystallise the complex with this mutant?

8) It may be useful to include a supplemental table with the primers used to clone regions of the different proteins in expression vectors as well as for the site directed mutagenesis.

9) Please, replace "ball and stick format" by "stick representation" or "stick format" in legends to Figures 1-5.

10) Supplementary Figure 2b. Please, be consistent in listing the Uniprot accession numbers and species.

Reviewer #3 (Remarks to the Author):

The authors have investigated the role of the conserved plakin linker domain in intermediate filament (IF) binding. This is a conserved domain that has been shown to have a role in IF binding, but the mechanism is unknown. Desmoplakin contains this domain in between the second and third plakin repeat domains, which are known to bind IFs, and some desmoplakin ARCV. Starting from an unpublished structural genomics structure of the periplakin linker, the authors show that an electropositive groove that links two plakin repeat-like elements is a binding site for vimentin, using both cellular colocalization and direct binding of wild type and mutated variants. They generate a model of the desmoplakin linker and show that it binds weakly to vimentin, and rationalize the effect of ARCV mutations. They also provide a model for the interaction of the linker with the IF coiled-coil. Overall the work should be of considerable interest to biologists working in the area of cytolinkers and intermediate filaments.

There are a number of concerns with the data and analysis that need to be addressed.

1. The statistical significance of differences between wild-type and mutant vimentin colocalization data in Figures 1 and 3 is not clear. There are no p values provided for the MOC graphs, nor the number of cells and pixels analyzed). There are many problems in interpreting MOC (see Alder et al 2010 *Cytometry* 77A: 733-42 or Dunn et al 2011 *Am J Physiol Cell Physiol* 300:C723-C742). At a minimum, they should show that their answer does not depend on the choice of correlation metric.

2. MST assays are very sensitive to protein aggregation. Apart from those studied by NMR, what is the validation that the linker or vimentin mutants are properly folded? Given that the curves don't achieve saturation, can they be sure that there isn't a non-specific signal here? The authors should provide at least some examples of the raw MST curves to convince the reader that the protein is well behaved. This is especially a concern for the desmoplakin E2495K/C2497R mutant.

3. The binding of the periplakin linker to vimentin is quite weak, which may reflect the multivalent interactions of these proteins. In the case of desmoplakin, the binding is extremely weak (which was also reported, albeit indirectly, in Ref. 2). In the latter case, could a role of the linker be to provide proper geometric positioning of the two flanking PRDs?

4. The analysis of the R2541K ARVC mutant (lines 273-277) is confusing. The authors note that both R and K can form a salt bridge with D2545. So what do they think is the actual effect of the mutation?

5. The modeling and mutational study of the periplakin raises the question of IF specificity. D176 is conserved, but E187 is not. The discussion implies that these charged residues are critical for IF binding generally, so a more careful discussion should be provided.

Response to referees:

Reviewer 1 Comment 1a: Analysis of the previously released crystal structure of periplakin linker (PDB ID 4Q28) is an important starting point to address that periplakin linker forms a positively charged groove where vimentin IF binds. However, this crystal structure forms a stable beta-sheet mediated by a hexa-histidine tag sequence and crystal contacts with neighboring molecules in the crystal lattice, which are not physiologically relevant. It is not clear whether periplakin linker still forms a bi-lobed module connected by long beta-strands as shown in Figure 1a after removal of hexa-histidine residues from the structure and isolation of single protomer from the crystal lattice. The authors also mention that this structure has crystallization artifacts but the crystal structure was used to model the linker domain without experimental verification. This structure is very critical for this manuscript because all model structures including linker-vimentin complex were generated using this structure as a template. Therefore, it is necessary to clarify that Figure 1a represents the structure of periplakin linker in solution. Do you have any experimental evidence that periplakin linker forms a central beta-sheet as shown in Figure 1a, such as CD spectrometric analysis?

The HSQC data for the periplakin (Fig. 1b) and desmoplakin (Fig. 4a) linker domains reveals well-dispersed peaks suggesting that each module adopts an alpha/beta fold. We have included secondary structure prediction for the desmoplakin linker based on consideration of the crystal structure, I-TASSER model, and NMR chemical shift data calculated using Talos+ (Supplementary Fig.2a). This data suggests that the desmoplakin linker in solution encompasses a central beta sheet and generally mirrors the periplakin linker crystal structure. Given the high degree of sequence similarity between the periplakin and desmoplakin linker we think that the periplakin module is likely to adopt a similar overall topology in solution (page 7, Results, paragraph 1). Importantly, in the revised study the I_TASSER derived periplakin linker model was used in generating complexes with vimentin fragments (Fig. 6 a-b, Supplementary Table 3). CD spectra of two different preparations of the periplakin linker show negative absorption at 218 nm, indicative of presence of beta strands (Supplementary Fig.2c), which is consistent with NMR analysis of the desmoplakin linker.

Reviewer 1 Comment 1b: Or, do you have any modeling data (of MD simulation) in the absence of hexahistidine residues and crystal contacts?

We have generated a model of the periplakin linker domain incorporating residues K1646-K1756 using I-TASSER and this demonstrates a bi-lobed structure connected by beta strands even in the absence of the hexahistidine tag. The I-TASSER method for protein structure prediction includes fragment-guided molecular dynamics simulation. This model is shown in Figure 1a and discussed in Results, page 7, paragraph 1.

Reviewer 1 Comment 1c: More importantly, Figure 1a and the secondary structure presentation in Figure S2(a) do not match each other. Based on Figure S2 (and PDB), R1655 is located at the N-

terminal region of the first beta-strand but it is shown near the C-terminal region of the first beta-strand in Figure 1a. Could you explain this?

The secondary structure presentation highlighted in Supplementary Fig.2 was an older version that was inadvertently incorporated in the submitted version. This has been corrected in the revised version and more importantly Figure 1a and the secondary structure presentation in Supplementary Fig.2a now match each other.

Reviewer 1 Comment 1d: What residues form the first beta-strand in Figure 1a? If hexa-histidine residues are included in the structure representation (in Figure 1a), that is very misleading. Only the periplakin linker residues should be presented. Also, please label the beta strands such as S1, S2, etc. in Figure 1a and Figure S2(a).

Figure 1a in the original submission represented the published crystal structure of the periplakin linker domain with the first beta strand corresponding to the His-tag residues. We appreciate the reviewers concerns and hence in the revised manuscript we include an I-TASSER derived molecular model of the periplakin linker domain (Figure 1a). This model now excludes the N-terminal His-tag but importantly does include periplakin N-terminal residues starting from K1646. The molecular model of periplakin clearly demonstrates a bilobed module connected by beta strands. A similar structure can also be derived using the molecular modelling program Phyre2. In addition secondary structure prediction programs such as JPred predict a beta strand at the N-terminus encompassing residues E1648-S1657 consistent with the I-TASSER derived periplakin linker model. Also, as requested by the reviewer, beta strands such as S1 and S2 have now been labelled in the revised figures (Fig.1a, Fig. 3a and Supplementary Fig.2a).

Reviewer 1 Comment 2a. Related to point #1, computational modeling constitutes an important part of this manuscript. The accuracy and reliability of model is critical for the follow-up experiments and conclusions. In plectin linker and desmoplakin linker models (Figures 2d and 3a, respectively), structural variations of PR2-like domain and Nt PR-like domain are observed; for example, two beta-strands are missing in PR2-like domains. In contrast, the central beta-strands are well maintained in all plakin linkers. Do you think crystallographic artifacts in the template structure are removed during modeling? Do these models match well with secondary structure prediction analysis?

We believe that plectin and desmoplakin linker domains demonstrate a similar secondary structural topology to the periplakin linker domain. The reason for the variation in the original figures was that the molecular visualization program PyMOL failed to assign some of the strands in the I-TASSER derived plectin and desmoplakin linker models. These have now been corrected in the revised figures (e.g. Fig.2d and Fig.3a).

Reviewer 1 Comment 2b: Also, model coordinate files in the supplementary are essential. It is hard to compare linker structures in the figures.

For the benefit of the reviewers we have included model coordinate files for the different linker structures in the form of appendices 1-3.

Reviewer 1 Comment 3a. It is more appropriate to move the first section of results (lines 118 to 138, “Structure of the periplakin linker reveals a distinctive positively charged groove”) to the introduction or to combine it with the second section of results because not enough original data is included in this section. Especially, structural analysis of periplakin linker (lines 121 to 125 and 128 to 132) was already published by Kang *et al.*, (2016), although they questioned the formation of a central beta-sheet structure in physiological conditions.

Some of the first section of the results has been moved to the introduction as requested. However as there is no mention of a basic groove in the Kang *et al.*, study (PLoS ONE 2016) we feel that this structural feature is sufficiently novel to merit description in the first section of the results.

Reviewer 1 Comment 3b. In lines 132-133, the authors say the conserved N-terminal residues that could normally form part of the structure but it is not clear which residues are mentioned here. And, do the authors mean that these residues are conserved in all plakin proteins? Please clarify this.

The crystal structure of the periplakin linker domain encompasses a hexahistidine tag at the N-terminus with residues K1646-L1654 that normally form part of the structure missing. These missing N-terminal residues are relatively conserved across plakin family members. We have clarified this point in the revised manuscript on the bottom of page 5.

Reviewer 1 Comment 4: The authors show that R2541K mutation of desmoplakin exhibited a reduced MOC, similar to K2463E/R2464E and explain this mutation at the molecular level using the desmoplakin linker model structure (lines 271-277). However, this structural analysis doesn't explain how R2541K mutation showed a similar effect to the double mutations in the basic groove (K2463E/R2464E). The authors only say that R2541K mutation still maintains the overall structural integrity, allowing partial co-localization with IFs (lines 274-277). Can the authors explain why R2541K shows a reduced MOC?

Based on our modelling we believe that the R2541K mutation can compensate for the interactions mediated by R2541 and maintain the structural integrity of this region. However, comparison of desmoplakin linker wild-type and R2541K HSQCs reveals several residues with significant chemical shift perturbations within the Nt PR-like motif (Supplementary Figure 4a). It is possible that these changes may adversely impact IF binding and explain the lower MOC relative to wild-type desmoplakin. A sentence to this

effect has been included in the revised manuscript (results - Vimentin co-localization is compromised by desmoplakin linker domain mutations section – paragraph 3).

Reviewer 1 Comment 5: The authors present the model structure of the periplakin linker-vimentin complex by using the HADDOCK program and validation with affinity measurements of charge reversal mutants. However, the experimental validation does not fully support the complex model, because MST data using the mutants only show that D176, E180, and E187 are important for the interaction with the periplakin linker, but do not specify the interacting residues of the periplakin linker. If the authors can show that the abolished affinity of D176K mutant of vimentin(ROD) is recovered by using R1713D(or E) mutant of the periplakin linker, that can support the model.

As suggested by the reviewer the periplakin linker module incorporating the R1713E mutant has been made and assessed for binding with D176K vimentin^{ROD}. The binding interaction between the periplakin linker R1713E and the D176K vimentin^{ROD} mutants is shown in Figure 6d, Supplementary Fig.12 and Supplementary Table 4, along with the single mutant and wild-type combinations. The R1713E periplakin linker bound to the vimentin D176K mutant with a slightly higher affinity than the wild type linker and rod domains. These results are discussed in the last paragraph of the Results section, Page 17, and support the validity of the model.

Reviewer 1 Comment 6: In Fig. 1b, Fig. 3b, please label each color.

The information has been provided in the figure legends.

Reviewer 1 Comment 7: In Fig. 6d, please include wild type data.

The wild-type data is now included in the Fig. 6c.

Reviewer 1 Comment 8: In Fig. S4 (c) and (d), it should be D2545, rather than D2541.

These figures have been altered as required.

Reviewer 1 Comment 9: In line 201, it should be “1694-1698”, rather than “1694-1699”.

This has been corrected in the revised manuscript.

Reviewer 1 Comment 10: In line 202, reference in parenthesis should be shown as number (22)

This has been corrected in the revised manuscript.

Reviewer 2 Comment 1: Figures 1b and 3b. The quantification of the co-localization experiments should be readdressed. Please, indicate the specific number of images per condition that were used to calculate the Manders' overlap coefficient and the number of independent experiments (or replicates) in the figure legends. A Tukey box plot showing the median and 25th and 75th percentiles for each distribution as well as the indication of the p-values is recommended. What are representing the error bars?

The methodology for calculating the Manders' overlap coefficient and the number of replicates has been clarified in the Methods section. As requested by the reviewer Tukey box plots have been used with p values provided in figure legends (Fig. 1d and Fig. 3c).

Reviewer 2 Comment 2: I have some concerns about the poor fit of the MST data for the mutations R1737E/K1741E in periplakin or the mutation E180K in vimentin (Figure 2a and Figure 6d). Could authors explain these behaviours? In addition, it may be helpful to have a supplementary table that includes the K_d values and r² for each independent experiment.

We have added Supplementary Tables 1, 2 and 4 which list the binding affinities, number of data sets (n) and the standard error of regression (S) for each individual data set and the merged data sets. The raw MST traces for all of the data sets are shown in Supplementary Figures 4, 5, 7, 10 and 12.

Our MST data on the E180K vimentin variant has been reassessed and clearly exhibits a higher affinity for the periplakin linker compared to wild-type vimentin. One possibility for this increase in affinity is that the lysine can form an ionic interaction with E1692 which borders the basic groove of the periplakin module. A statement to this effect has been added to the revised text (page 17, last paragraph of results). The fit of the R1737E/K1741E data returns a higher standard error of regression but the slightly increased binding affinity seen in the MST assays (Fig.2a) is also seen in the cellular assays (Fig.1c) and is discussed on page 10, top paragraph.

Review 1 Comment 3: The authors claim that an electrostatic component is key for the interaction between periplakin and vimentin. This could be further supported with MST experiments at different salt concentrations. Could the salt concentration of the reaction have any effect on the K_ds?

This is excellent point raised by the reviewer. We have found that the salt concentration does have an effect on the K_D. At low salt concentration (10 mM NaCl) the binding affinity is increased 2.5 fold (31 μM vs 71 μM). The binding affinity at 50 mM is only slightly lower relative to assays performed at 150 mM NaCl. These results indicate that electrostatic forces are driving the interaction between the periplakin linker and vimentin. We have included several sentences (results - Mutations in the basic groove compromise periplakin's

interaction with vimentin section – paragraph 3), an additional figure (Supplementary Fig.5) and a table (Supplementary Table 2) to address these points in the revised study.

Reviewer 2 Comment 4: The PDB doi link (<http://dx.doi.org/10.2210/pdb4Q28/pdb>) for the structure with PDB code 4Q28 should be mentioned and cited.

The relevant link has now been added in the revised text (page 7, Results, first paragraph).

Reviewer 2 Comment 5: In Supplementary Figure 1a, the addition of the domain organisation for BPAG1e and plectin would help the readers to better understand the manuscript.

The domain organization for BPAG1e and plectin has now been included in the revised manuscript in Supplementary Fig. 1a.

Reviewer 2 Comment 6: Periplakin-vimentin complex computational modelling. In model 3 (the most supported model) the periplakin mutations R1689, R1655 and R1713 were identified as key-residues for the interaction. In my opinion, every single mutant should also be tested in order to confirm the inhibitory effect and validate the complex model. Moreover, these results, together with those for vimentin D176 and E187, should be mentioned in the abstract, since they contribute to pinpoint the impact of the manuscript. Since these two last single mutations completely abolish the interaction, a possible structural impact of these variants should also be analysed. Are these variants well folded when a lysine is introduced? Circular dichroism, SAXS, etc. could be useful to answer this question.

We have chosen to mutate closely positioned charged residues in pairs to maximize the potential effects on vimentin binding. This is an approach we have used successfully before (Fogl, 2016). We have constructed the R1713E mutant which is described below.

We have added a sentence to the abstract that reflects the importance of vimentin residues D176 and E187 in binding to the periplakin linker domain.

The proton NMR spectra of all the vimentin mutants used in the MST assays are shown in Supplementary Fig. 11. These spectra demonstrate that all of the mutants are folded and comparable to the wild-type protein. In addition we have shown that the periplakin linker R1713E mutant is able to bind to vimentin D176K even though wild-type linker does not interact with this mutant.

Reviewer 2 Comment 7: Need to discuss why vimentin E180K mutant stabilises the interaction with periplakin. Have authors tried to crystallise the complex with this mutant?

Please see response to reviewer 2 comment 2 for why the E180K mutant stabilizes the interaction with periplakin. We have not attempted crystallization of the complex with this mutant as it is beyond the scope of this study.

Reviewer 2 Comment 8: It may be useful to include a supplemental table with the primers used to clone regions of the different proteins in expression vectors as well as for the site directed mutagenesis.

We have now included Supplementary Table 5 that lists the primers used to construct the periplakin and desmoplakin constructs. Mutants were produced using the QuikChange Lightning site directed mutagenesis kit using mutagenic primers designed according to the manufacturers' instructions. This is highlighted in the Methods section.

Reviewer 2 Comment 9: Please, replace “ball and stick format” by “stick representation” or “stick format” in legends to Figures 1-5.

As requested by the reviewer the ball and stick format has been replaced with stick representation.

Reviewer 2 Comment 10: Supplementary Figure 2b. Please, be consistent in listing the Uniprot accession numbers and species.

These have been corrected in the revised manuscript.

Reviewer 3 Comment 1: The statistical significance of differences between wild-type and mutant vimentin colocalization data in Figures 1 and 3 is not clear. There are no p values provided for the MOC graphs, nor the number of cells and pixels analyzed). There are many problems in interpreting MOC (see Alder et al 2010 Cytometry 77A: 733-42 or Dunn et al 2011 Am J Physiol Cell Physiol 300:C723-C742). At a minimum, they should show that their answer does not depend on the choice of correlation metric.

The methodology has been clarified in the methods section and p values are now provided in figure legends. Manders' coefficient is the most appropriate metric for our analysis for the reasons stated in the manuscript. Using an inappropriate metric to analyse the data for comparison purposes would not we believe be a useful exercise.

Reviewer 3 Comment 2: MST assays are very sensitive to protein aggregation. Apart from those

studied by NMR, what is the validation that the linker or vimentin mutants are properly folded? Given that the curves don't achieve saturation, can they be sure that there isn't a non-specific signal here? The authors should provide at least some examples of the raw MST curves to convince the reader that the protein is well behaved. This is especially a concern for the desmoplakin E2495K/C2497R mutant.

The requested binding experiment data are now included in the supplementary material. Importantly, there was no aggregation detected in the traces. The linker domains at higher concentrations (mM) form higher order structures (dimers) and the protein will aggregate with time. To alleviate these effects linker proteins were concentrated immediately prior to analysis and the levels were kept to a maximum of 800 μ M in the binding reactions to avoid non-specific interactions. The vimentin^{ROD} domains did not show any adsorption or aggregation in the MST assays and the proton NMR spectra of the wild-type and mutants demonstrates correctly folded proteins (Supplementary Fig. 11). A statement about the lack aggregation has been added to the MST method and the NMR characterization has been included in the vimentin purification protocol.

Reviewer 3 Comment 3: The binding of the periplakin linker to vimentin is quite weak, which may reflect the multivalent interactions of these proteins. In the case of desmoplakin, the binding is extremely weak (which was also reported, albeit indirectly, in Ref. 2). In the latter case, could a role of the linker be to provide proper geometric positioning of the two flanking PRDs?

This is an interesting point and as a result we have added a sentence in the discussion (page 19-20) to address this.

Reviewer 3 Comment 4: The analysis of the R2541K ARVC mutant (lines 273-277) is confusing. The authors note that both R and K can form a salt bridge with D2545. So what do they think is the actual effect of the mutation?

See response to Reviewer 1 Comment 4

Reviewer 3 Comment 5: The modeling and mutational study of the periplakin raises the question of IF specificity. D176 is conserved, but E187 is not. The discussion implies that these charged residues are critical for IF binding generally, so a more careful discussion should be provided.

The discussion (page18, second paragraph) has been modified in the revised text to reflect this point.

REVIEWERS' COMMENTS:

Reviewer #1 (Remarks to the Author):

In the revised manuscript, the authors have addressed the reviewer's concerns.

The authors provide convincing evidence showing that the positively charged groove of the periplakin linker interacts with the acidic residues of vimentin.

In the last section of the result, the authors show that D167K vimentin mutant binds to R1713E periplakin mutant, to support vimentin-periplakin linker complex model. However, some other mutant data do not match well with this complex model 2. For example, E172K and E180K mutants have the enhanced binding affinity for the periplakin linker.

Line 388-389 : Two mutants, E172K and E229K exhibited a moderate increase in linker binding affinity whilst one, E180K, exhibited a larger increase in affinity.

However, the authors say,

Line 393-395 : Together this supports complex model 2 wherein vimentin residues E172 and D176 from coil 1B are recognized by periplakin R1655 and R1713, while vimentin's E180 contact periplakin residues R1689 and K1714

I would recommend modifying this sentence to match the mutation data.

* Minor point

1) Surface potential representation of the periplakin linker is repeated in Fig. 1a and Fig. 2b. Basically, they are the same figures. Please take one of them.

2) In Fig. 2d, only K4275 and R4277 side chains are shown but the authors say,
Line 704-705: Residues 4274 RKRR 4277 are shown (stick format).

Reviewer #2 (Remarks to the Author):

Although all specific comments from the first revision round were accepted, I still have one major concern that should be addressed:

In response to comment 1, authors clarified in the Methods section that generally at least 5 images, containing 1-2 transfected cells were collected from each co-localization experiment (repeated 2-3 times). This number of analyzed cells is unusually low in this kind of experiments. Could authors justify why they did not use a higher number? Moreover, the indication of the specific number of analyzed images for each experiment in the corresponding figure legend is also recommended.

Other minor considerations:

1) Please, add the y-axis title in Figure 3c and correct the following references to figures:

- Fig.1b by Fig.1d in line 218.

- Fig.4 by Fig.5b in lines 451 and 456.

- Fig.4d by Fig.4a in line 473.

- Supplementary Fig.4 by Supplementary Fig.6a in line 484.

2) Please, indicate the different NaCl concentrations used for the evaluation of the salt effect in the Methods section.

Reviewer #3 (Remarks to the Author):

The authors have responded carefully to the concerns raised in the initial reviews. I have a few minor points below, but I would ask the authors to consider, in their Discussion, the role of non-electrostatic interactions with IFs as well, rather than giving the impression that everything seen here is driven by electrostatics. There are two driving concerns here. First, the entire analysis depends on charge reversal mutations, rather than charge neutralization. The former are clearly deleterious and could exert long-range repulsive effects, but this does not say that the compatible positive-negative pairs contribute substantially to the binding energy. There is an extensive literature showing, for example, that surface charge-charge interactions may not contribute much free energy to binding due to competition with bound water or ions (e.g. see Sauer's classic work on surface charge stabilization of protein folding). Second, the new salt dependence study (Fig. S5) shows a very modest effect on affinity – a twofold change which is energetically very small. It is not supportable to claim, on line 224, that this change indicates that “electrostatic attraction plays a major role”.

On a perhaps more “philosophical” point, given that for practical reasons the experiments do not achieve saturation, the distinction between no binding and non-specific electrostatic effects of the mutants are not completely obvious.

Minor:

Line 133: in response to reviewer 2, reference 20 is cited as the reference for the periplakin linker (PDB 4Q28) crystal structure. This is incorrect - that reference merely used that structure for modeling; the structure appears to have come from an unpublished structural genomics study.

The “explanation” for the ARVC R2541K desmoplakin mutation is not really an explanation. Apparently the change does alter the structure but no binding data are provided to assess what is going on.

Response to referees:

Reviewer #1

Line 393-395 : Together this supports complex model 2 wherein vimentin residues E172 and D176 from coil 1B are recognized by periplakin R1655 and R1713, while vimentin's E180 contact periplakin residues R1689 and K1714

I would recommend modifying this sentence to match the mutation data.

We have modified the text as requested

- 1) Surface potential representation of the periplakin linker is repeated in Fig. 1a and Fig. 2b. Basically, they are the same figures. Please take one of them.
- 2) In Fig. 2d, only K4275 and R4277 side chains are shown but the authors say, Line 704-705: Residues 4274 RKRR 4277 are shown (stick format).

As requested by the reviewer we have removed Fig.2b. In Fig.2d (Fig. 2c in the revised manuscript) the R4274 and R4276 side chains have also been shown in stick format.

Reviewer #2

In response to comment 1, authors clarified in the Methods section that generally at least 5 images, containing 1-2 transfected cells were collected from each co-localization experiment (repeated 2-3 times). This number of analyzed cells is unusually low in this kind of experiments. Could authors justify why they did not use a higher number? Moreover, the indication of the specific number of analyzed images for each experiment in the corresponding figure legend is also recommended.

We have revised the methods "For quantification of immunofluorescent microscopy at least 5 fields were examined for each experiment, with each field containing 2-4 transfected cells. z-stacks (slice thickness 0.7 μm) were taken for each field and overlap coefficients calculated for each individual z-stack. An average overlap coefficient was then calculated for each experiment and each experiment was repeated 2-3 times.". We have also added the following to the legends for Figs 1 and 3 "At least 5 fields of view were analysed for each experiment. z-stacks were taken for each field and overlap

coefficients calculated for each individual z-stack. An average overlap coefficient was then calculated for each experiment and each experiment was repeated 2-3 times.” Although additional images were collected, 5 representative images of each condition were analysed in detail and are reported, as is consistent with previous publications in Nature journals.

1) Please, add the y-axis title in Figure 3c and correct the following references to figures:

- Fig.1b by Fig.1d in line 218.

- Fig.4 by Fig.5b in lines 451 and 456.

- Fig.4d by Fig.4a in line 473.

- Supplementary Fig.4 by Supplementary Fig.6a in line 484.

2) Please, indicate the different NaCl concentrations used for the evaluation of the salt effect in the Methods section.

All of these changes have been made in the revised manuscript.

Reviewer #3

The authors have responded carefully to the concerns raised in the initial reviews. I have a few minor points below, but I would ask the authors to consider, in their Discussion, the role of non-electrostatic interactions with IFs as well, rather than giving the impression that everything seen here is driven by electrostatics. There are two driving concerns here. First, the entire analysis depends on charge reversal mutations, rather than charge neutralization. The former are clearly deleterious and could exert long-range repulsive effects, but this does not say that the compatible positive-negative pairs contribute substantially to the binding energy. There is an extensive literature showing, for example, that surface charge-charge interactions may not contribute much free energy to binding due to competition with bound water or ions (e.g. see Sauer’s classic work on surface charge stabilization of protein folding). Second, the new salt dependence study (Fig. S5) shows a very modest effect on affinity – a twofold change which is energetically very small. It is not supportable to claim, on line 224, that this change indicates that “electrostatic attraction plays a major role”.

We now state in the discussion (paragraph 2) that ‘this is not simply a matter of basic character and forces other than electrostatic interactions, including steric fit and hydrophobic interactions, may also be in play’. We have removed the

word 'major' from the sentence above.

Line 133: in response to reviewer 2, reference 20 is cited as the reference for the periplakin linker (PDB 4Q28) crystal structure. This is incorrect - that reference merely used that structure for modeling; the structure appears to have come from an unpublished structural genomics study.

We accept this point by the reviewer and have changed the text to 'The crystal structure of the periplakin linker was determined by the Northeast Structural Genomics Consortium (PDB entry 4Q28) and briefly described by Weiss and colleagues²⁰'.

The "explanation" for the ARVC R2541K desmoplakin mutation is not really an explanation. Apparently the change does alter the structure but no binding data are provided to assess what is going on.

We have rewritten the text regarding R2541K. We now state 'The majority of residues exhibiting the largest chemical shift perturbations were restricted to the Nt PR-like element (**Supplementary Fig.6b**). It is possible that these changes adversely impact IF binding, explaining the lower MOC relative to wild-type desmoplakin (**Fig.3c**)'. We accept that we cannot go further in the absence of binding or other data.